# Transcription start sites experience a high influx of heritable variants fueled by early development

Miguel Cortés Guzmán [1,2,6], David Castellano [1,5,6],
Clàudia Serrano Colomé[1,2,6], Vladimir Seplyarskiy[3,4] & Donate Weghorn [1,2] ✉

Mutations drive evolution and genetic diversity, with the most consequential mutations occurring in coding exons and regulatory regions. However, the impact of transcription on germline mutagenesis remains poorly understood. Here, we identify a mutational hotspot at transcription start sites (TSSs) in the human germline, spanning several hundred base pairs in both directions. Notably, the hotspot is absent in de novo mutation data. We reconcile this by showing that TSS mutations are significantly enriched with early mosaic variants, many of which are excluded from de novo mutation calls, indicating that the hotspot partly arises during early embryogenesis. We associate the TSS mutational hotspot with divergent transcription, RNA polymerase II stalling, R-loops, and mitotic—but not meiotic—double-strand breaks, suggesting a recombination-independent mechanism distinct from known processes. Our findings are reinforced by mutational signature analysis, which highlights alternative double-strand break repair and transcription-associated mutagenesis. These insights reveal a germline mutational phenomenon with evolutionary and biomedical implications, particularly affecting genes linked to cancer and developmental phenotypes.

The mutation rate in human cells is influenced by DNA sequence composition and epigenetic factors[1–3]. Although germ cells and somatic tissue cells share many of these factors, there are fundamental differences in the observed distribution and nature of mutations, particularly due to exogenous and disease-related mutational processes active in the soma[4]. In addition to replication time, which is one of the most important determinants of mutation rate, transcription and the directionality of its effects are also debated[5,6]. Transcription affects mutation rate in at least two important ways: (1) via the joint effects of transcription-associated mutagenesis (TAM) and transcription-coupled repair (TCR) on the transcribed sequence itself and (2) via the effects of transcription factor (TF) binding in the regulatory sequence elements[7].

The balance of TAM and TCR on expressed genes is a topic of great interest. In the soma, gene expression is often thought to correlate negatively with observed mutation density, likely due to a dominant role of TCR, especially in tumours with a strong influx of mutation-inducing DNA damage[2,8]. In the germline, the consensus is not as clear. A recent study reported an overall negative correlation between transcription and mutation density, similar to that observed in the soma[9,10]. However, this effect was not observed in another study[4] and reversed when some potential confounding factors were taken into account[11], consistent with previous reports[12,13]. Hence, it is currently uncertain how large the actual impact of transcription on mutation rates in the germline is when all important covariates of mutation rate are taken into account. Regarding the influence of TFs, it

[1]Centre for Genomic Regulation (CRG), The Barcelona Institute of Science and Technology, Dr. Aiguader 88, Barcelona, Spain. [2]Universitat Pompeu Fabra (UPF), Barcelona, Spain. [3]Division of Genetics, Brigham and Women's Hospital, Harvard Medical School, Boston, MA, USA. [4]Department of Biomedical Informatics, Harvard Medical School, Boston, MA, USA. [5]Present address: University of Arizona, Tucson, AZ, USA. [6]These authors contributed equally: Miguel Cortés Guzmán, David Castellano, Clàudia Serrano Colomé. ✉e-mail: dweghorn@crg.eu

has been shown that promoter-proximal TF binding sites in the soma exhibit an increased mutation density, especially in melanomas and binding sites belonging to CTCF[14–16]. This effect is likely due to a combination of limited DNA repair and, in particular, increased DNA damage[17,18]. Interestingly, Perera & Poulos et al. (2016) found that TF binding alone cannot be the sole reason for the increased mutagenicity at DNAse I hypersensitive sites (DHSs) and speculated that transcription initiation must play an important role[16]. In the germline, an excess of mutations around testis-active promoter-proximal TF binding sites, especially for T > G mutations, was recently found[19,20].

To shed more light on these open questions, we investigated the effects of transcription on mutation rate by examining the variability of observed mutation density in transcribed regions and their genomic neighbourhood. Using extremely rare variants (ERVs) from two large human whole-genome sequencing cohorts, we uncovered a pronounced mutational hotspot of non(CpG > TpG) mutations near the transcription start site (TSS) in the human germline, extending several hundred base pairs in both directions. Surprisingly, the hotspot is not detectable in de novo mutations (DNMs) from family sequencing data. We show that this apparent discrepancy is resolved by a significant enrichment of the TSS with early mosaic mutations, many of which are removed in family sequencing mutation calling pipelines, unveiling early development as a key factor in TSS mutagenesis. Consistently, analysis of the mechanistic factors driving TSS mutagenesis showed no association with meiotic double-strand breaks, but instead revealed that the hotspot is associated with divergent transcription, stalling of RNA polymerase II (RNAP II), R-loops and mitotic double-strand breaks. Mutational signature decomposition further corroborates the role of mosaic variants occurring during mitotic cell divisions and suggests involvement of non-canonical double-strand break repair, transcription-dependent mutational processes, and a process typically associated with clustered mutations that arise in the female human germline.

## Results

### Transcription start sites of protein-coding genes show an excess of extremely rare polymorphisms.
To quantify the direct and indirect effects of transcription on the mutation rate in the human genome, we analysed four sets of mutations: Extremely rare variants that segregate in the human population obtained from (1) gnomAD[21] and (2) the UK Biobank (UKBB, www.ukbiobank.ac.uk; sample allele frequency < 0.01%), (3) de novo mutations from family sequencing data (Methods) and (4) somatic mutations from cancer genomes[22]. Like other studies, we considered ERVs a proxy for neutral germline variation, as they are only slightly affected by the effects of selection[23,24]. We excluded hypermutable tumours from the pan-cancer cohort, as they exhibit greater sequence context dependencies than our model considers[25,26]. Furthermore, we subdivided the mutations into the two main classes of non(CpG > TpG) and CpG > TpG mutations, as the latter have a 1-2 orders of magnitude higher average mutation rate due to methylation-dependent mutagenicity[27].

For each set of mutations, we then calculated the relative mutation density in non-overlapping genomic windows of 100 or 1000 base pairs (bp) in length. We anchored our analysis at the transcription start site (TSS) and transcription termination site (TTS) of 14763 non-overlapping protein-coding genes and analysed up to 50 kilo-base-pairs (kb) upstream and downstream of the TSS and TTS. Our measure of relative mutation density per window is the ratio between the observed and expected number of mutations averaged over genes, $\mu$. The expected number of mutations takes into account the differences in mutation propensity due to DNA sequence composition. It is based on a 5-mer sequence context model, which is calculated separately for the four different mutation data sets and the two mutation classes from the respective genome-wide average mutation probabilities. Because we treated complementary mutations separately and

orientated them along the direction of transcription, this model also accounts for the effects of transcription strand bias[24]. Since we found that both gene length and distance to other genes are positively correlated with $\mu$, likely due to their correlation with replication time[28], we also defined a second measure of sequence-corrected mutation density, $\mu'$, for visualisation. This measure was designed to capture the variability of mutation density at length scales ≤ 150 kb in and around transcribed regions that is neither caused by DNA sequence composition nor by the larger regional variability in mutation rate.

The most remarkable feature of the relative mutation density in transcribed and neighbouring untranscribed regions that we found is a pronounced mutational hotspot at the TSS for non(CpG > TpG) ERVs, Fig. 1a–b. This localised excess reaches up to 14% at the 1-kb scale. Zooming in to 100 bp, the relative density of ERVs in the first 100 transcribed sites is increased by up to 35%, with the total regional mutation excess extending several hundreds of base pairs in both directions. There is also evidence of a hotspot of cancer mutations at the TSS, but it is much narrower (≈ 300 bp in total) and shifted upstream towards untranscribed regions, Fig. 1c, consistent with previous results[16]. Stratification by cancer type showed that this signal is driven by several tumour types, including bladder, breast, oesophageal, head and neck, lung, lymphoid, ovarian, pancreatic and gastric cancers (Supplementary Figs. 1–2). In lymphoid tumours, the mutational excess is restricted to the first 2 kb downstream of the TSS, compatible with somatic hypermutation in the variable domains of immunoglobulin genes[29], whereas in all other cancer types with a hotspot the mutational excess has an upstream component. Notably, liver tumours show the opposite pattern, with a clear deficit of mutations especially in the first 1-2 kb downstream.

Previous work has shown that TF binding in cancer and in the germline can lead to locally increased mutation at promoter DHSs, suggesting that TSS hypermutability may be related to transcription[16,20]. Since transcription in the human genome is not restricted to protein-coding genes, we applied the same methodology as for protein-coding genes to three types of non-protein-coding genes: 3991 divergent lncRNAs, 8454 intergenic lncRNAs and 1660 pseudogenes. The promoters of intergenic lncRNAs are usually embedded in enhancers, whereas the promoters of divergent lncRNAs are shared with the promoters of protein-coding genes and transcription occurs upstream and in the opposite direction[30]. Our analysis revealed an even larger TSS hotspot of non(CpG > TpG) mutations on divergent lncRNAs than for protein-coding genes, both for ERVs (47% excess for gnomAD and UKBB) and for somatic cancer mutations (49% excess) at 100 bp (Supplementary Figs. 3–8). Conversely, mutations on pseudogenes and on intergenic lncRNAs showed only a slight excess ( < 11%) near the TSS at the 100-bp length scale for ERVs and no significant excess for the pan-cancer data set. Although divergent lncRNAs are expected to be transcribed more frequently than intergenic lncRNAs, these results may suggest that, in addition to gene expression itself, bidirectional transcription is mechanistically related to the TSS-proximal mutation hotspot.

In contrast to non(CpG > TpG) mutations, we found a deficit of CpG > TpG mutations in all mutation sets near the TSS at the 1-kb level, Fig. 1d–f, as expected due to a lower mutation rate of unmethylated CpG islands in the vicinity of the TSSs of expressed genes[31]. Interestingly, against the background of this general deficit, ERVs show a small relative excess of CpG > TpG variants in the first 100 bp downstream of the TSS. Finally, for all mutation sets and classes, we found on average a lower $\mu'$ in transcribed than in neighbouring untranscribed regions (see Supplementary Fig. 9 for results showing $\mu$), which can be explained in large part by the strong entanglement of transcription and replication landscapes, especially in the soma[32] (Supplementary Fig. 10). To test the effects of negative selection on our results, we repeated the same analyses without exons and conserved non-coding sequence elements. If strong negative selection removed mutations at

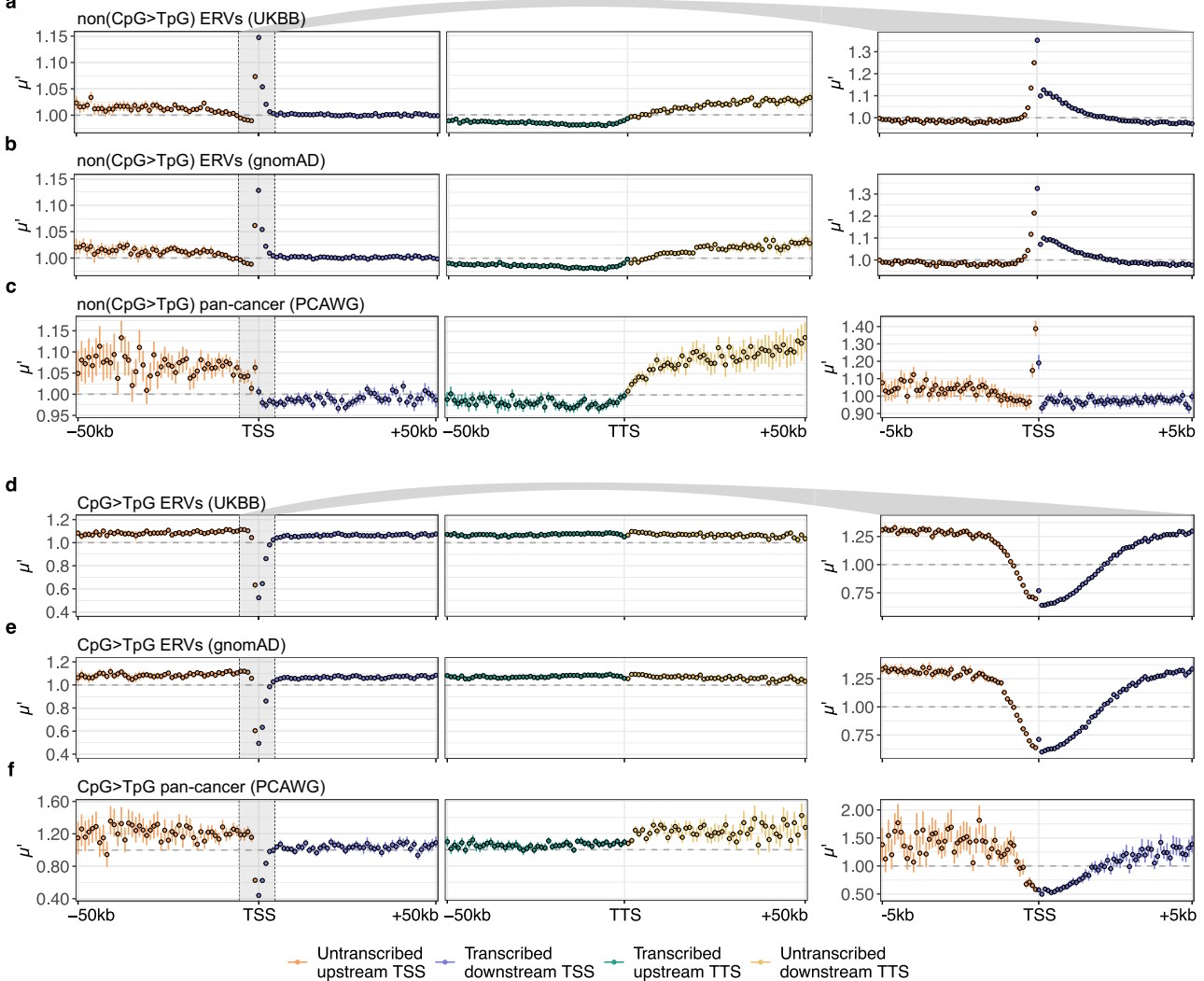

**Fig. 1 | Extremely rare variant and pan-cancer mutation density around the transcription start and termination site.** Average number of mutations across 14763 protein-coding genes divided by the expectation based on the 5-mer sequence context and by the gene-level mutation density, $\mu\prime$, upstream and downstream of the TSS (orange and blue, respectively) and upstream and downstream of the TTS (green and yellow, respectively). Each dot represents $\mu\prime$ in a 1-kb window (left) or 100-bp window (right). Error bars represent the 90% confidence intervals across 100 bootstrap replicates. Non(CpG > TpG) mutations using (**a**) UKBB ERVs, (**b**) gnomAD ERVs and (**c**) pan-cancer somatic mutations. **d–f** CpG > TpG mutations using the same data sets. Source data are provided as a Source Data file.

the TSS, we would expect to observe an even more pronounced hotspot when we exclude conserved sites. However, the TSS hotspot for ERVs is reduced after removal of exons and conserved sequences (Supplementary Figs. 11–12), consistent with the mutation rate being primarily a function of distance from the TSS.

**Early mosaic mutations are significantly overrepresented near the transcription start site.** A more direct indicator for the germline mutation rate than ERVs are de novo mutations (DNMs). To analyse DNMs, we assembled a large meta-cohort of ten published data sets[33–42]. Surprisingly, DNMs did not show a significant excess of non(CpG > TpG) mutations near the TSS, which could not only be explained by the smaller sample size, especially downstream, Fig. 2a. At the same time, we did recover the expected CpG > TpG deficit in TSS DNMs, suggesting that something specific to non(CpG > TpG) variants was causing the difference between DNMs and ERVs, Fig. 2b. We hypothesised that a possible explanation could lie in the filtering applied to de novo mutation calls from family sequencing data. First, mutations that match common variants segregating in the human population are often filtered out, but this should remove a negligible

number of variants[34]. A more important factor could be that, inherent to the DNM data acquisition process from family sequencing, early mosaic variants that occurred in the parents are inadvertently removed during mutation calling, Fig. 2c.

Mosaic variants have been found to contribute to several of the germline mutational signature components identified in Seplyarskiy & Soldatov et al.[24], which represent independent mutational processes involved in ERV mutagenesis. Therefore, we next decomposed the ERV mutational profile near the TSSs of protein-coding genes into these germline signature components, Fig. 2d. We found that the relative enrichment of a component near the TSS compared to the gene body correlates significantly with the enrichment of that component with mosaic variants compared to de novo variants, as identified in Seplyarskiy & Soldatov et al. ($p = 0.006$, Pearson), Fig. 2e. Accordingly, three of the four components significantly enriched in mosaic variants are increased 1.5- to 4.8-fold near the TSS (components 5/6, 7 and 11), supporting the hypothesis that the TSS is particularly influenced by somatic mutagenesis during early development.

Given these indications that mosaic variants might strongly contribute to the TSS mutational hotspot, we next compiled a data set of

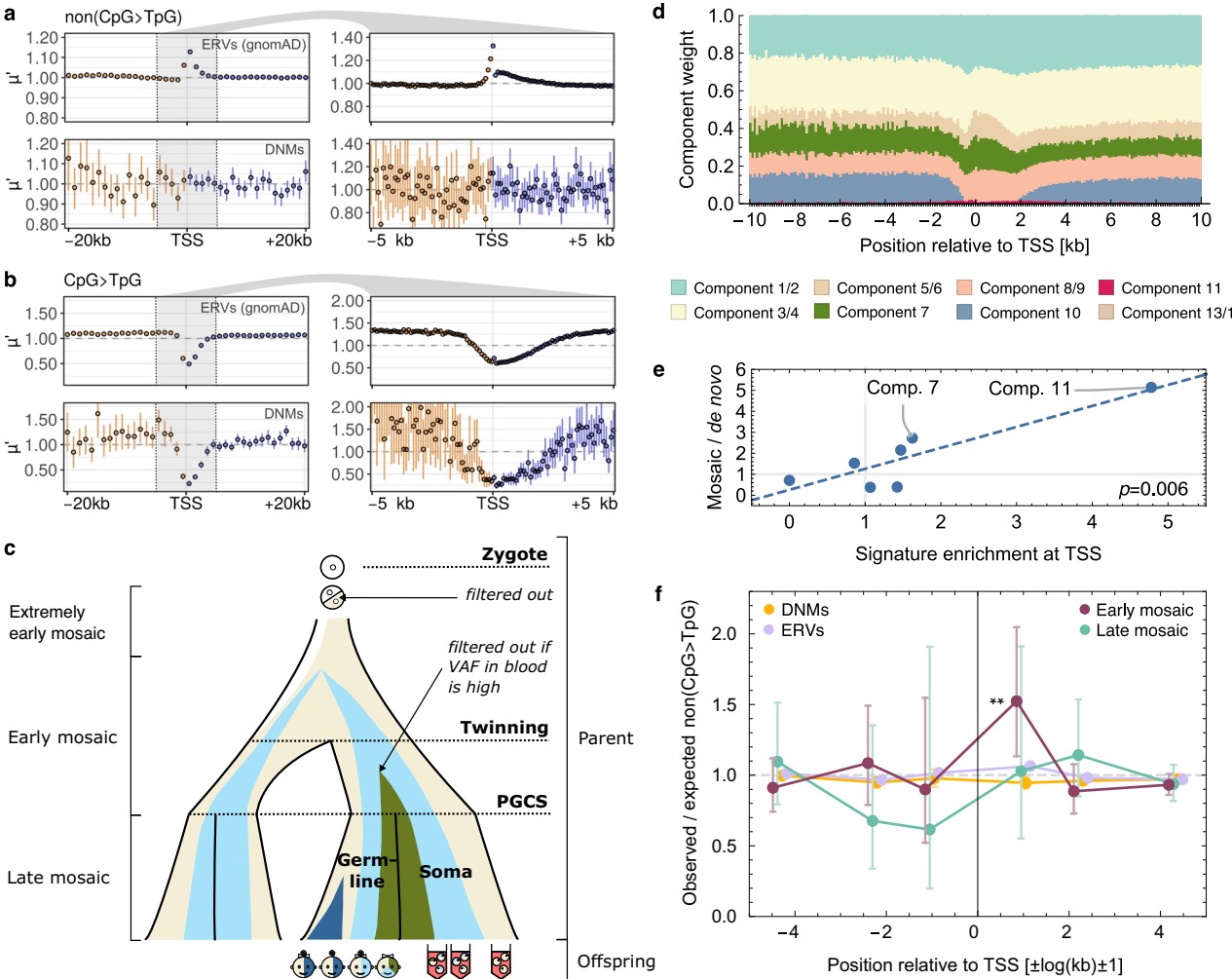

**Fig. 2 | Early mosaic non(CpG > TpG) variants are enriched at the TSS.** Relative mutation density $\mu'$ upstream (orange) and downstream (blue) of the TSS for gnomAD ERVs (top panels) and DNMs (bottom panels) in 1-kb windows (left) and 100-bp windows (right) across 14763 protein-coding genes: (**a**) non(CpG > TpG) mutations, (**b**) CpG > TpG mutations. Error bars indicate the 90% confidence intervals for the mean across 100 bootstrap replicates. **c** Illustration of mosaic mutation timing and filtering in standard family sequencing data sets, inspired by Fig. 1 in Jonsson et al.[33]. PGCS=primordial germ cell specification. **d** Decomposition of ERVs in 100-bp bins into germline mutational signature components[24]. **e** Enrichment of mutational signature components with mosaic mutations (defined

in ref. 24) as a function of the ratio of the component weight in the first and the 15th 1-kb bin downstream of the TSS. **f** Ratio of observed and expected number of non(CpG > TpG) early mosaic (maroon), late mosaic (turquoise), de novo (yellow) and extremely rare SNP (lilac) variants around TSSs of 14763 protein-coding genes, averaged in bin brackets ± [1], ± [2,5] and ± [6,50] kb. Expected number based on a mononucleotide model. Data are presented as Poisson means with error bars denoting 95% confidence intervals derived from standard errors of the mean. **\*\****p* = 0.005 is the unadjusted *p*-value from a two-sided *z*-test. Source data are provided as a Source Data file.

mosaic mutations from 11 published studies[33–40,43–45], separating into early and late mosaic variants. We found a significant excess of 52% in the density of early mosaic mutations immediately downstream of the TSS ($p = 0.005$), Fig. 2f, suggesting that the difference between the 14% excess in downstream ERV density and the 4% excess observed in DNMs is indeed due to mosaic variants. At the same time, the observed ERV excess in the 1 kb upstream of the TSS (7%) is within the bootstrap intervals of the DNM data, indicating that the two data sets are compatible in this genomic region, consistent with no significant deviation of mosaic variant density from the expectation in this bin. Taken together, these results suggest that the TSS is highly enriched with mosaic mutations and their removal explains the absence of the mutation hotspot downstream of the TSS in family sequencing experiments.

**The TSS germline mutational hotspot is associated with measures of mitotic double-strand breaks, divergent transcription, RNA polymerase II stalling and R-loops.** Next, we aimed to identify which

genomic and epigenetic features were the drivers of the germline mutational hotspot at the TSS. To address this question, we performed a multiple negative binomial regression analysis of the sequence-corrected ERV mutation density, $\mu_{tb}$, across all protein-coding genes $t$ at the 1-kb window level using 38 feature variables, explicitly modelling the dependence of each feature on the position relative to the TSS (i.e., on the bin ID $b$) with interaction terms (Methods).

Reassuringly, when considering only the genome-wide average effects of each feature, our regression framework recovered the expected positive associations of the sequence-corrected non(CpG > TpG) mutation density with meiotic crossover sites[46], recombination rate[23] and maternal chromosomes[24] as well as negative associations with nucleosome occupancy[47] and replication time[23] (note that early replication corresponds to large replication time values), Supplementary Fig. 13. Similarly, the two strongest positive predictors of mutations on transcribed sequences in the pan-cancer data set are oncogene and tumour suppressor gene identity, in line with their mutational excess due to selection[26,48].

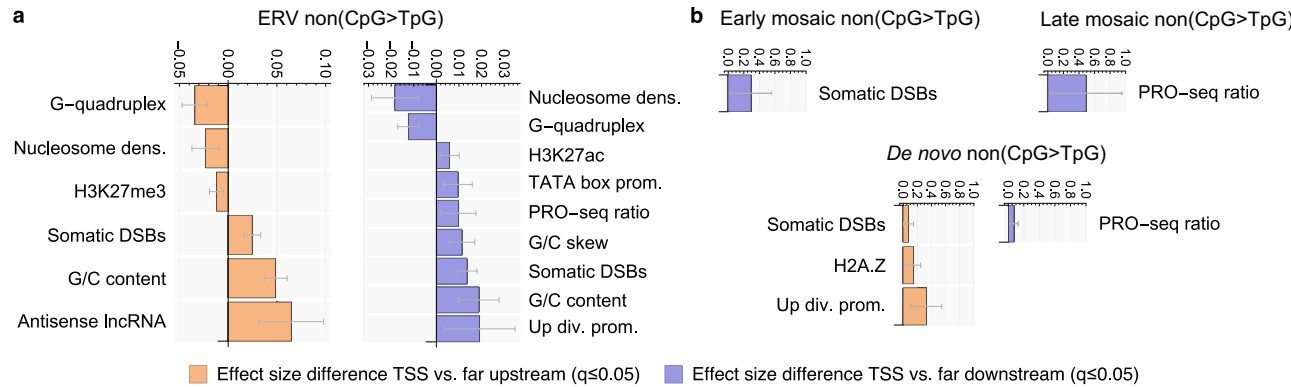

**Fig. 3 | Determinants of germline mutation density around the transcription start site. a** Difference in standardised multiple negative binomial regression coefficient of regressor × bin interaction between bin brackets [1] and [6,50] kb upstream (orange) and downstream (blue) of the TSS for non(CpG > TpG) mutations. All significant terms from multiple-testing-adjusted two-sided z-tests replicating across UKBB and gnomAD ERVs that also have a significant total effect size at the TSS in at least one of the two data sets are shown ($q \leq 0.05$, both regressions fitted to 226879 genomic windows). Error bars denote multiple-testing-adjusted 95% confidence intervals derived from coefficient standard errors using single-step intervals correction[130]. dens.=density, div. prom.=divergent promoter, DSBs=double-strand breaks. **b** Same as (**a**), but using single negative binomial regression with interaction terms for early mosaic, late mosaic and de novo variants. Regressions for DSBs, H2A.Z and Up div. prom. were fitted using 309659 windows and for PRO-seq ratio 249147 windows. Source data are provided as a Source Data file.

To explicitly capture the effect of each feature on mutation density specifically at the TSS, we then compared the coefficient of the interaction term at the TSS with the one far away from it, taking into account the covariance structure between the coefficients. We found that two features were significantly associated (adjusted p-value $q \leq 0.05$) with increased ERV density in the first 1 kb on both sides of the TSS: (1) annotated sites of somatic double-strand breaks (DSBs)[49] and (2) G/C content, Fig. 3a. Five additional features correlated significantly positively with ERV mutagenicity 1 kb downstream of the TSS: (3) H3K27ac, a histone mark found at the TSSs of actively transcribed genes[50], (4) TATA box promoter, (5) the ratio of PRO-seq read density at the TSS and further downstream, a measure of RNAP II stalling, (6) G/C skew, a measure of the probability of R-loop (i.e., transcription-associated DNA-RNA hybrid) formation[51], and (7) divergent transcription, defined based on GRO-cap data[52]. This last finding was supported by the strongest predictor of mutational excess 1 kb upstream of the TSS: (8) co-localisation of the protein-coding gene with an upstream antisense lncRNA gene, an alternative measure of divergent transcription. These findings replicated in both the gnomAD and the UKBB ERV data set.

The associations with H3K27ac and divergent transcription confirm the notion that both transcription itself and its bi-directionality independently increase germline mutation density at the TSS, consistent with the large mutational excess at divergent lncRNA promoters (Supplementary Fig. 3). R-loops and RNAP II stalling consolidate the link of the mutational hotspot to transcription, possibly capturing related phenomena, as both have been linked to genomic instability and DNA damage[53,54]. Potentially the most remarkable finding is the strong association of the ERV hotspot in the first 1 kb on both sides of the TSS with somatic double-strand breaks ($q \in [0, 10^{-2}]$ across UKBB and gnomAD). Since these DSB estimates are based on measurements in a healthy somatic cell line[49] and a recent study found mutational hotspots in the vicinity of putative sites of meiotic DSBs[46], we also included the annotations of those sites in our feature list (PRDM9). While we indeed found a significant positive genome-wide effect of PRDM9 on non(CpG > TpG) mutation probability in both ERV data sets across transcribed and untranscribed regions ($\hat{\beta} \approx 0.004$, $q < 10^{-2}$, Supplementary Fig. 13, Supplementary Data 1), the signal was not statistically significantly associated with the TSS (Supplementary Fig. 14, Supplementary Data 2), suggesting that the association of the TSS mutational hotspot with somatic DSBs has a different aetiology to PRDM9-mediated events.

We next aimed to test if mosaic variants recapitulated the associations with genomic and epigenetic features observed for ERVs by regressing separately on each of the 38 feature variables and their interaction with bin ID. Despite the much smaller sizes of these mutation data sets (early mosaic: 0.002%, late mosaic: 0.001%), this identified two of the seven features that were also significantly positively associated with ERV mutation density in the first 1 kb downstream of the TSS: somatic double-strand breaks (early mosaic) and PRO-seq read density ratio (our proxy for RNAP II stalling; late mosaic), Fig. 3b. To further query the mutagenic role of transcription-associated phenomena at the TSS during early human development, we compared the correlation between gene expression and the mutation excess at the TSS across human developmental stages. We found that the expression-associated mutagenesis intensifies around the major transcriptional state shift between the 4-cell and 8-cell stages and continues at a comparable level into the differentiation of the trophectoderm (Supplementary Fig. 15).

DNMs confirmed the significant association with RNAP II stalling (PRO-seq ratio, $q < 0.05$) downstream. This effect may be partly influenced by mosaic variants, as most late mosaic variants in our DNM data set could not be excluded due to the limited availability of studies involving families with multiple offspring. We also note that almost all of the DNM data sets we used include the child's mosaic variants, as these are not filtered out in standard family sequencing DNM calling. Upstream of the TSS, we recovered a significant association of DNMs with DSBs and divergent transcription as we had already observed for ERVs, supporting an important role for these factors in promoter mutagenesis.

**The mutation hotspot at the TSS is associated with extraordinary somatic mutational processes.** Given its unusual nature, we hypothesised that the mutational hotspot near the TSS is due to mutational processes that diverge from background processes ubiquitously active in the human germline and soma[4]. Therefore, we decomposed the mutational profile in each window around the TSSs of protein-coding genes into COSMIC mutational signatures for single base substitutions (SBS)[55], quantifying both the relative and the absolute number of mutations attributable to each signature (corrected for the target size), Fig. 4a–c. Although these mutational signatures were derived from cancer sequencing data, we hypothesised that there might be some overlap and informative similarities with germline mutational processes beyond the background signatures SBS1 and SBS5.

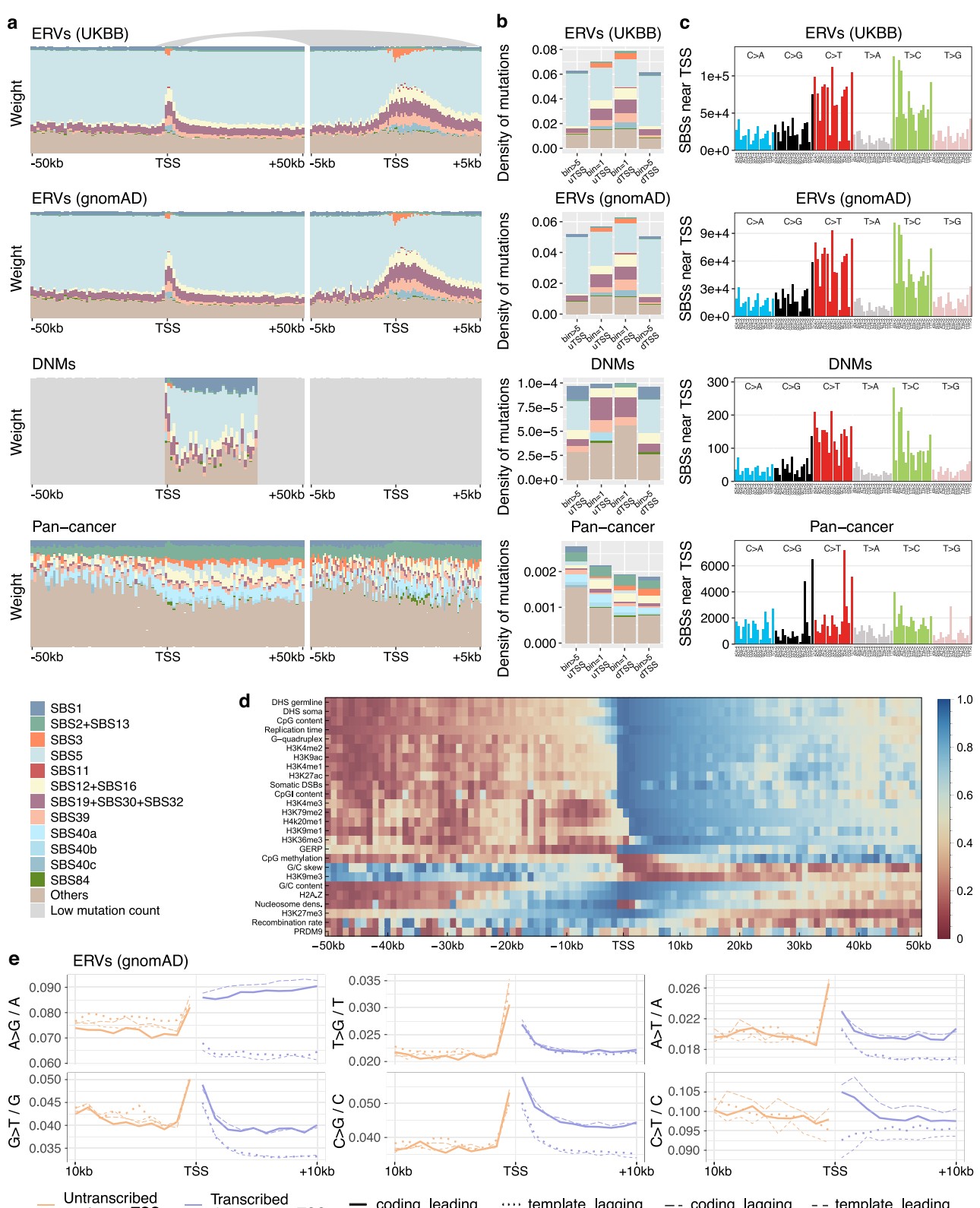

**Fig. 4 | Somatic mutational signatures and transcription strand biases affecting the TSS mutational hotspot. a** Somatic mutational signature decomposition in 1-kb (left) and 100-bp (right) windows around the TSSs of protein-coding genes, for ERVs (UKBB and gnomAD), DNMs and PCAWG pan-cancer mutations (top to bottom). Bins with < 1000 mutations pooled across genes are marked in grey. **b** Mutation density stratified by signature in bins upstream (uTSS) and downstream (dTSS) of the TSS, shown for the 1-kb bins directly neighbouring the TSS (bin=1) and for mutations pooled across bins 6 to 50 kb (bin > 5). **c** Mutational profile in the five

bins around the TSS [-2,3] kb after subtraction of SBS1, SBS5 and SBS40b,c.
**d** Relative contribution (quantiles) of non-gene-level genomic and epigenetic features used in the regression analyses as a function of position relative to the TSS. Note that early corresponds to high and late to low values of the replication time measure. **e** ERV non(CpG > TpG) mononucleotide mutation density, stratified by transcription strand (coding or template) and replication strand (leading or lagging) in 1-kb windows. Source data are provided as a Source Data file.

The decomposition of the ERV profiles revealed four mutational signatures that occur exclusively in the region around the TSS: SBS3, SBS11, SBS40b and SBS40c. The first of these signatures, SBS3, has been associated with deficiencies in homologous recombination repair (HRR) of double-strand breaks in cancer tumours[56]. A likely candidate for alternative repair of such DSBs is the polymerase theta-mediated end joining (TMEJ) pathway[57]. We recently found that indels associated with this repair pathway co-localise with SBS3 point mutations as well as evidence linking SBS40 (whose subcomponents are SBS40b and SBS40c) to TMEJ during aborted HR or stalled replication[58]. Taken together, the somatic mutational signatures strongly suggest non-canonical DSB repair impacting the TSS germline mutational hotspot, consistent with the significant association of TSS hypermutability with somatic DSBs (Fig. 3).

We also found several other non-background mutational signatures with increasing contribution closer to the TSS: SBS12, SBS16, SBS19, SBS30 and SBS39, all of which are also present near the TSS in the DNM data set. Of these, the largest proportion of mutations near the TSS originates from SBS39 (-10%). This signature is dominated by C > G mutations and shows a high similarity to a germline mutational signature component associated with maternal de novo mutation clusters (component 8/9 in Seplyarskiy & Soldatov et al.[24]). To test the de facto association between SBS39 and component 8/9, we stratified the mutations by chromosomes enriched in maternal mutation clusters (maternal chromosomes 8, 9, 15 and 16[24,34,36]) and found that SBS39 mutation density on these chromosomes is systematically increased in all three germline mutation data sets (Supplementary Fig. 16). While signature SBS39 is present on transcribed sequences further downstream (and more so than in untranscribed regions, in agreement with[24]), the much larger number of mutations associated with this signature near the TSS suggests that the underlying mutational process is particularly influenced by TSS-proximal events.

As it has been hypothesised that maternal clustered mutations are associated with meiotic crossover events[34,36], the latter could be the driving mechanism behind SBS39 enrichment at the TSS, even though we found no interaction of PRDM9 or recombination rate with the TSS mutation excess (Fig. 3). However, both recombination rate and PRDM9 levels (measuring the DNA occupancy of the homology search protein DMC1[46]) are relatively low in the 1-kb bins neighbouring the TSS compared to sites further upstream or downstream, Fig. 4d. This suggests that meiotic double-strand breaks, in addition to their negligible effect on mutational excess near the TSS, also have a small effect on the relative weights of mutational signatures near the TSS.

In addition to C > G, the TSS mutational profile showed pronounced contributions from C > T and T > C mutations after subtraction of background processes, Fig. 4c. T > C variants are in agreement with the increased weight of signatures SBS12 and SBS16 near the TSS. In cancer, these signatures are primarily observed in liver tumours and co-occur so frequently that they contaminate each other as well as the background process SBS5[59], suggesting a more fundamental role in mutagenesis. Notably, SBS5 is relatively less active near the TSS. Both SBS12 and SBS16 exhibit a large transcription strand bias, suggesting a differential effect of TAM and TCR[60,61]. To link this to germline mutations, we calculated the transcription and replication strand biases for all mononucleotide mutation types, indeed showing strong biases for T > C ERVs, Fig. 4e. In contrast to T > C variants, the large contribution of C > T mutations to the background-corrected mutational profile near the TSS has no obvious counterparts among somatic mutational signatures. Instead, we hypothesise that this accounts for the identification of several signatures that together approximate the observed C > T pattern in Fig. 4c, including SBS11, SBS19, SBS30 and SBS32. In cancer tumours, the characteristic APOBEC-induced C > G and C > T mutation patterns in TCA and TCT contexts (corresponding to SBS2 and SBS13) dominate the mutational spectrum around the TSS, which is consistent with previous reports[62].

Besides T > C, we also found large transcription strand biases for all other mutation types except T > G (see Supplementary Fig. 17 for cancer mutations), while the mutation rate is accelerated both upstream and downstream for almost all mutation types near the TSS (see Supplementary Fig. 18 for the 100-bp scale). In a previous report, the large TSS-proximal excess of C > T mutations on the coding strand was associated with damaged ssDNA during R-loop formation[60]. However, we observe that in addition to an increase on the coding strand, the C > T mutation density is decreased on the template strand, which requires an alternative explanation. Finally, we also found an association with the direction of replication, particularly for C > T and T > C mutations (Fig. 4e), implying that both transcription and replication co-operate to generate these mutation types[63].

Regarding the background process SBS1, this CpG > TpG-associated signature contributes between 10–20% of mutations in DNMs, while its weight in ERVs is reduced to a few percent, highlighting the recurrence of CpG > TpG mutations and their subsequent undercounting in the ERV class[64]. At the same time, all four data sets in Fig. 4a show the expected depletion of SBS1 mutations in the 2-4 kbs around the TSS due to demethylation of CpG islands (Fig. 1d-f), which coincide with the promoter regions of 60–70% of genes.

**The TSS mutational hotspot impacts disease-associated genes and is eroded by purifying selection.** ERVs serve as proxies for germline mutational processes unaffected by selection. When repeating our analysis with higher-frequency variants, we found that the TSS hotspot of the evolutionarily younger mutations is not reflected in older polymorphisms, Fig. 5a. This is consistent with direct and background selection acting on the hotspot[65] and can explain why it was not detected in substitution data[31]. Quantifying direct selection yielded a dN/dS ratio of 0.35 for common exonic SNPs and a significant depletion of non-synonymous ERVs (dN/dS ≈ 0.82), indicating that variants with allele frequencies up to 0.01% (i.e., < 14 and < 30 copies in gnomAD and UKBB, respectively) were already subject to measurable purifying selection, Supplementary Fig. 19. Note that these dN/dS estimates are likely underestimating the true strength of selection, as we are considering segregating variants[66]. In contrast, DNMs showed a slight dN/dS increase above 1, reflecting the overrepresentation of diseased individuals in this cohort[67]. GC-biased gene conversion (BGC), which favours C/G over A/T nucleotides, may further contribute to hotspot erosion[68]. We found the TSS hotspot is strongest for mutations indifferent to BGC (A < > T, C < > G, 48% excess) or favoured by BGC (AT > CG, 39%) and weakest for BGC-unfavoured mutations (CG > AT, 24%), while BGC-unfavoured mutations showed the greatest reduction at TSSs among common SNPs. Taken together, these observations support both the roles of purifying selection and BGC in hotspot erosion during evolution, Fig. 5b.

Next, we asked whether the purifying selection that suppresses the mutational excess over longer evolutionary timescales could be linked to specific diseases. Calculating the relative strength of the TSS hotspot for all disease gene sets listed in the Human Phenotype Ontology[69], we found highly significant associations ($p_{adj} < 10^{-50}$) with over 20 neoplasms and carcinomas (mean Jaccard index JI=0.22), decreased mitochondrial activity (two phenotypes; JI=0.52), seven neurological phenotypes (JI=0.18) and defective limb development (three phenotypes; JI=0.11), Fig. 5c. This suggests that studies based on single-generation family sequencing data may miss predisposing variants associated with mental health disorders, but also other developmental defects and cancer due to the inadvertent filtering of mosaic TSS variants[39,45]. At the biological pathway level, we found significant enrichment of genes associated with DNA repair, protein degradation, RNA processing and translation ($p_{adj} < 10^{-2}$), rendering these pathways particularly vulnerable to germline mutations, Fig. 5d.

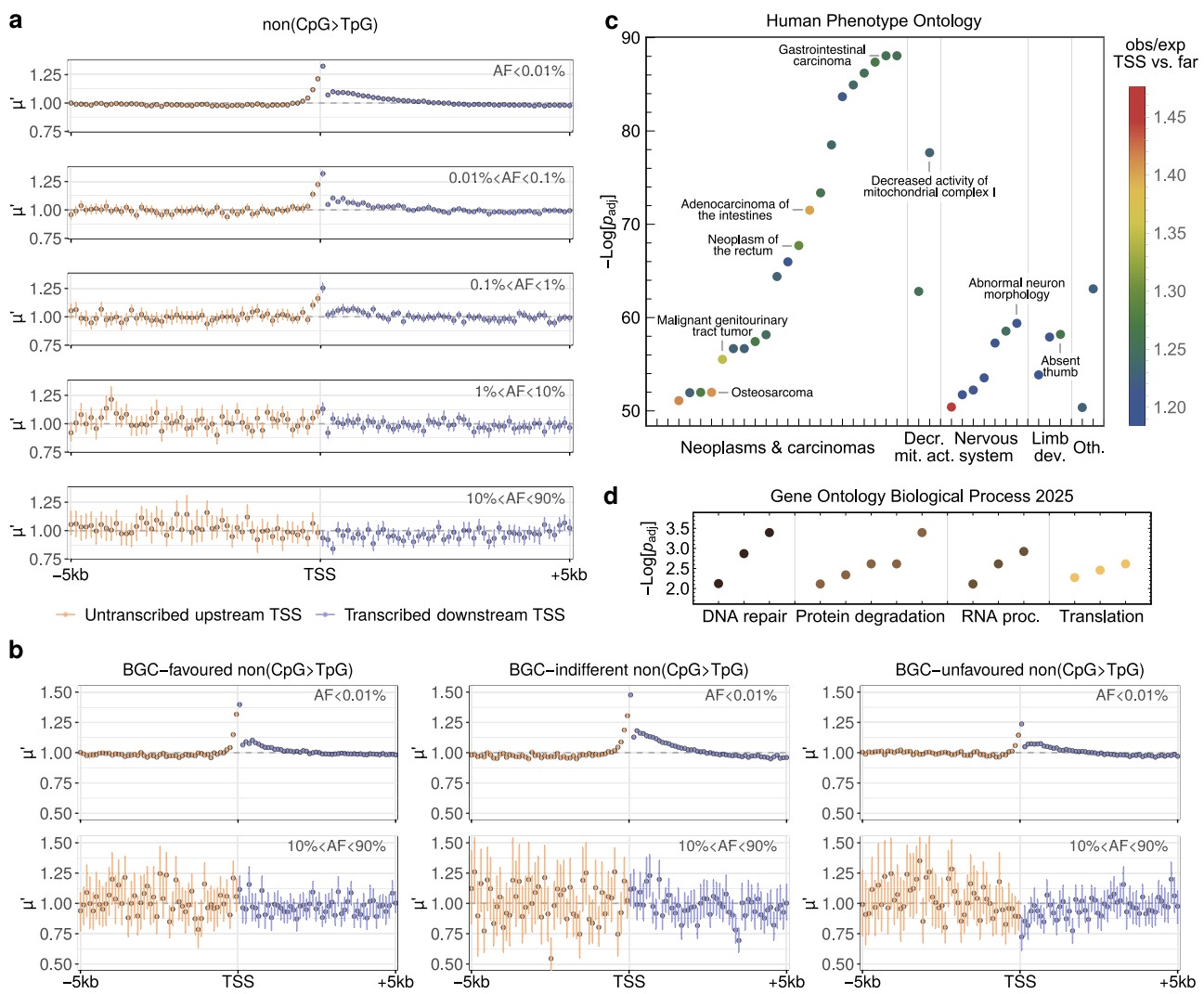

**Fig. 5 | Impact of the TSS mutational hotspot on disease phenotypes and erosion due to selection and biased gene conversion. a** Regional variability in $\mu'$ across 14763 protein-coding genes for gnomAD non(CpG > TpG) SNPs in 100-bp bins around the TSS for allele frequencies below 0.01% (ERVs), in [0.01%, 0.1%], in [0.1%, 1%], in [1%, 10%] and in [10%, 90%], top to bottom. **b** Regional variability in $\mu'$ stratified by biased gene conversion (BGC) with favoured, indifferent and unfavoured mutations (left to right) and by allele frequency (top to bottom). **c** Significance of the ratio of the observed TSS-proximal mutational excess compared to TSS-distal regions in Human Phenotype Ontology (HPO) phenotype-associated gene sets ($p_{adj} < 10^{-50}$; ratio > 1.1, colour code), grouped by superterms. Bonferroni-corrected adjusted $p$-values obtained from a two-sided binomial test. **d** Gene Ontology (GO) Biological Process term enrichment with genes showing a significant ratio of TSS-proximal to TSS-distal mutational excess in a two-sided binomial test (Benjamini-Hochberg BH $p_{adj} < 0.01$). Enrichment obtained using EnrichR[141]. Decr. mit. act.=Decreased mitochondrial activity, dev.=development, Oth.=Other, proc.=processing. Error bars in (**a**) and (**b**) indicate the 90% confidence intervals for the mean across 100 bootstrap replicates. Source data are provided as a Source Data file.

**Gene expression correlates positively with mutation density in pancancer data and the germline when accounting for epigenetic and genomic confounders.** Finally, we used our genome-wide window-based negative binomial regression model, controlling for 38 possible covariates of mutation rate, to test the global effects of expression on mutation density. A recent study found an overall negative correlation between transcription and germline mutagenesis, except in a small group of highly expressed genes where mutation rates rise again[9,10]. However, a later re-analysis came to opposite conclusions[11], which in turn are consistent with earlier observations[12,13]. In the soma, a significant negative correlation between transcription level and mutation rate is usually assumed[2]. Here, using the expression data from Xia et al.[9] we found a significant positive association between $\mu_{tb}$ and gene expression for non(CpG > TpG) on transcribed sequences in both ERV data sets ($\hat{\bar{\beta}} \approx 0.004$, $q < 10^{-15}$) and a partially significant negative correlation for CpG > TpG ERVs ($\hat{\bar{\beta}} \approx -0.003$, $q \in [0.004, 0.2]$), Fig. 6a-b and Supplementary Fig. 13. In cancer, replication time is the most important predictor of mutation density in all mutation types and regions ($\hat{\bar{\beta}} \approx -0.238$, $q < 10^{-15}$), which is consistent with previous reports[2,13]. Interestingly, correcting for this and the other features, we found a positive correlation with gene expression in our non-hypermutable cancer tumours for all mutation types ($\hat{\bar{\beta}} \approx 0.006$, $q \in [10^{-8}, 1]$), Fig. 6c-d. While this observation may be different for some cancer types when considered in isolation, it suggests that the reported apparent negative effect of gene expression on somatic mutation density is primarily driven by other covariates of mutation rate, mainly replication time[32,70] (Supplementary Fig. 10).

## Discussion

Transcription and its associated processes are pivotal in shaping mutation rates. In somatic cells, it has been well-established that promoter regions upstream of the transcription start site exhibit elevated mutation rates in many cancers, which mechanistically has been explained by the impairment of nucleotide excision repair and

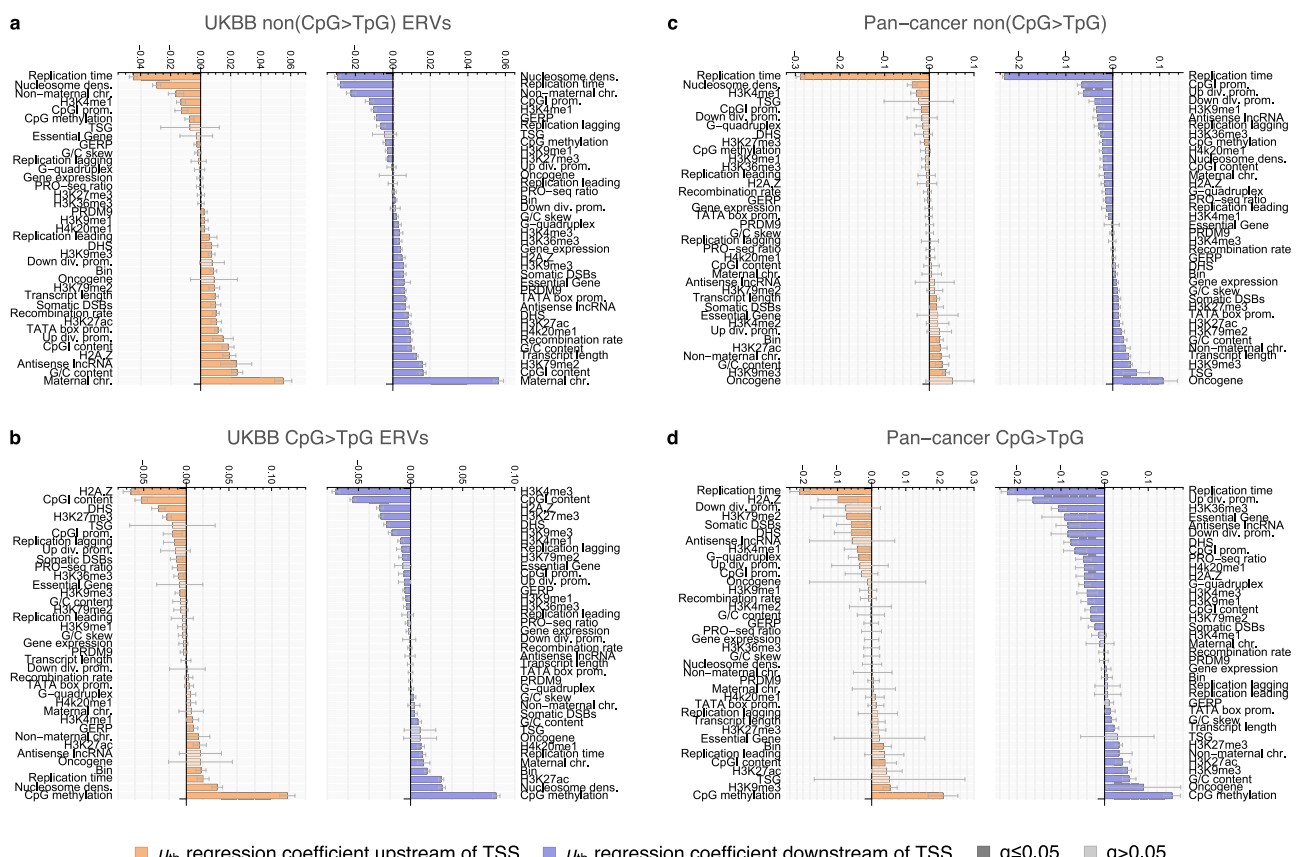

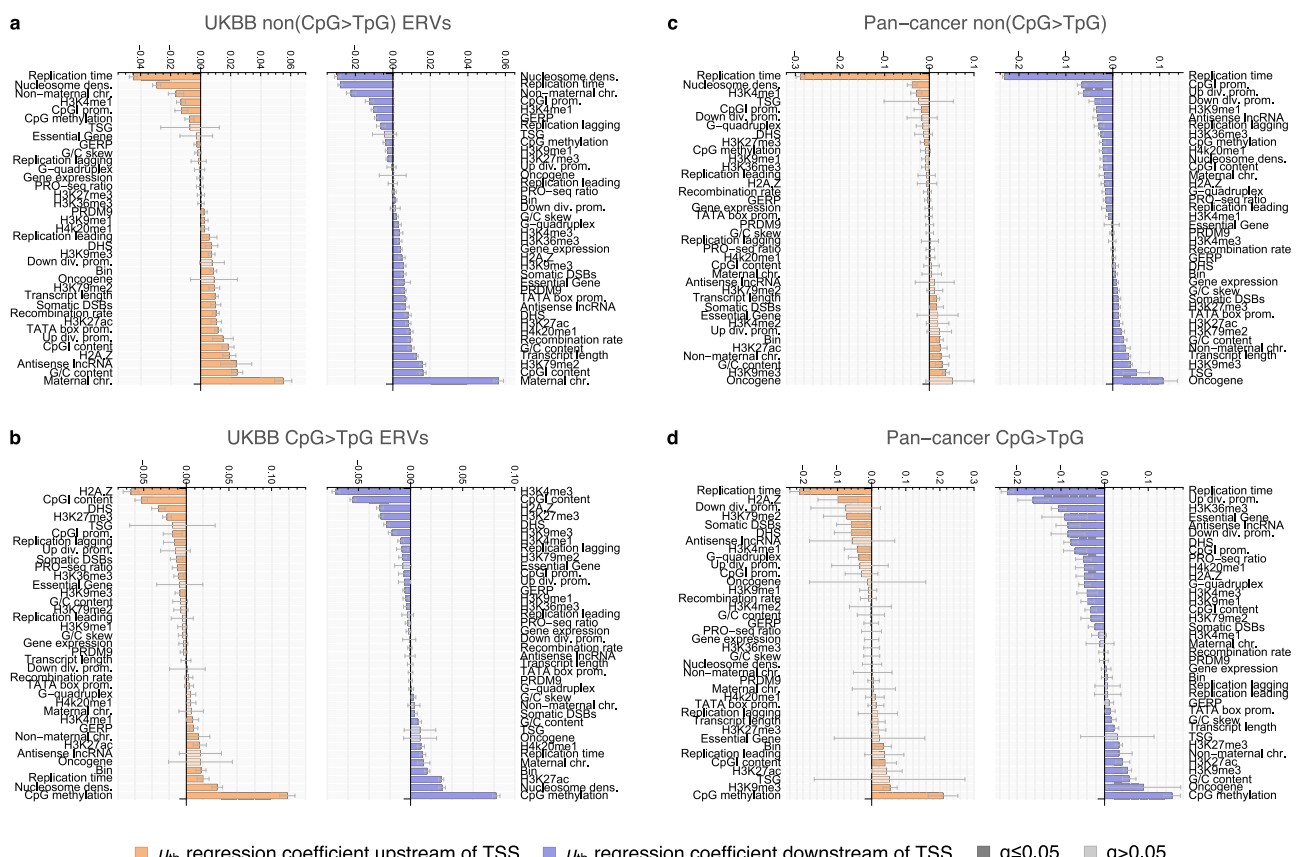

**Fig. 6 | Germline and somatic mutation rate covariates.** Standardised coefficients from multiple negative binomial regression of $\mu_{tb}$ across all protein-coding genes and 1-kb windows upstream of the TSS (orange) and downstream of the TSS (blue). **a** UKBB non(CpG > TpG) ERVs (fitted on 218819 windows downstream and 51966 upstream), (**b**) UKBB CpG > TpG ERVs (fitted on 211328 windows downstream and 48913 upstream), (**c**) pan-cancer non(CpG > TpG) variants (fitted on 228306 windows downstream and 54686 upstream), (**d**) pan-cancer CpG > TpG variants (fitted on 220350 windows downstream and 51419 upstream). Note that CpG > TpG ERVs are affected by mutation recurrence, so some associations may be distorted[64]. Further, early corresponds to high and late to low values of the replication time measure. Error bars show multiple-testing-adjusted standard-error-derived 95% confidence intervals of regression coefficients. Dark shading indicates an adjusted *p*-value $q \leq 0.05$ from a two-sided z-test. Source data are provided as a Source Data file.

increased DNA damage during transcription factor binding[15–17]. Here, we demonstrated the existence of a distinct mutational hotspot in the human germline, spanning several hundred nucleotides upstream and downstream of the TSS. Despite variations in ancestry among cohorts, we replicated these findings in two independent data sets, comprising 70,000 individuals (gnomAD) and approximately 150,000 individuals (UKBB).

Surprisingly, de novo mutations did not show a significant excess near the TSS. Beyond the smaller sample size and reduced statistical power of the DNM data set, we found that this is due to the systematic and inadvertent exclusion of early mosaic variants from family sequencing data, by observing a 52% excess ($p = 0.005$) of early mosaic variants within the first 1 kb downstream of the TSS. Mosaic variants have long been recognized as crucial contributors to germline mutational burden, with lower-bound estimates of their proportion ranging from 4 to 26%[34,35,71–73]. Our findings suggest that they strongly cluster around the TSS and, hence, that cohort studies relying on de novo mutations for disease associations may miss key regulatory and exonic mosaic variants at the TSS. We found that this particularly affects neoplasms, metabolic diseases, and neuronal and other developmental disorders. Similarly, a lack of knowledge about the mutational hotspot at the TSS may distort estimates of the strength of negative selection. A recent study that used ERVs to estimate the fraction of ultra-selected variants encountered an unexpected and implausible negative fraction of strongly deleterious variants at the TSS and traced its origin to the natural excess of mutations in this region[74]. Our

findings provide a framework to correct the mutational models underlying such studies, empowering more accurate inferences of negative selection in promoters, 5′-UTRs, and exonic regions[75,76].

Multiple independent mechanistic processes appear to drive the TSS hotspot in the germline. In contrast to our pan-cancer dataset, divergent transcription and mitotic double-strand breaks (DSBs) emerged as the dominant factors for ERVs upstream of the TSS. For the greater mutation excess downstream, also directly transcription-related mechanisms such as RNAP-II stalling and R-loop formation play a significant role. The involvement of mitotic DSBs and RNAP-II stalling also in mosaic downstream TSS mutagenesis, along with the observed correlation between the TSS hotspot and early developmental gene expression, point to a tentative timeline for TSS-related mutagenesis during development. Mutational signature analysis further supported these mechanisms, highlighting four key mutational processes: (1) T > C mutations linked to transcription (SBS12 and SBS16, known from liver tumours); (2) C > G mutations associated with transcription and complex crossovers in the female germline (SBS39); (3) C > T mutations of unknown origin (SBS11 and others); and (4) defective double-strand break repair (SBS3, SBS40b, and SBS40c, all mutation types).

Our results align with evidence of mutational excess at transcription factor binding sites of genes highly expressed in the male germline[20]. They also complement recent studies detailing mutational excess near recombination-associated PRDM9 binding sites[46] and constitutive replication origins[77], both associated with double-strand

breaks and TMEJ. Moreover, recent insights reveal connections between SBS3 and imperfect DSB repair with non-canonical recombination attempts[36], while meiotic DSB hotspots lacking PRDM9 binding fail to undergo homologue-guided repair[78]. These findings underscore the critical yet underexplored role of non-canonical DSB repair in germline mutagenesis. Our study places this process, alongside transcription, at the centre of TSS mutagenesis.

## Methods

### Definition of genomic windows
**Processing of genomic positions of interest.** We obtained FANTOM5 gene annotations from the ZENBU genome browser (https://fantom.gsc.riken.jp/zenbu/gLyphs/index.html)[79,80]. This resource provides robust locations of transcription start sites (TSSs) since annotations were created by associating peaks of Cap Analysis of Gene Expression (CAGE) to 5′ ends of genes annotated across different genomic sources[81,82]. There are 21069 protein-coding genes, 8187 divergent long non-coding RNAs (lncRNAs), 13105 intergenic lncRNAs, 4510 pseudogenes, and 12239 genes of other types in the FANTOM5 robust gene track. Genomic positions of interest were defined as a maximum of 50 kilo-base-pairs (kb) upstream and 50 kb downstream of both the TSS and transcription termination site (TTS). The analysis was limited to the gene length or the first/last 50 kb if the gene was shorter or longer than 50 kb, respectively. We binned these genomic positions of interest into non-overlapping windows of up to 1 kb or 100 base pairs (bp), taking as a starting point the TSS or TTS of each gene in the autosomes.

**Transcription, mappability, and liftover filters.** We excluded sites from the windows defined above that lay in untranscribed upstream TSS or downstream TTS regions that could actually be transcribed as part of other genes different from the one considered in each case. If untranscribed upstream or downstream positions from different neighbouring genes overlapped, we assigned the position to the nearest TSS or TTS. Hence, no position is represented more than once in our analysis and all positions are labelled as a function of distance to its closest TSS/TTS gene annotation. Sites transcribed as part of more than one gene, regardless of the transcription strand orientation (same or opposite), were excluded from the analysis.

We subtracted hard-to-map regions from the genomic windows in favour of using only sites present in uniquely mappable regions, based on the CRG-36mers alignability filter obtained from https://genome.ucsc.edu/cgi-bin/hgTrackUi?db=hg19&g=wgEncodeMapability[83]. These regions can promote the calling of artificial mutations and boost the counts of recurrent mutations across samples[83].

Since we worked with some mutation data sets originally provided in the hg38 genome assembly, in order to ensure a comparable window target size of mutation across data sets after using `liftover` to convert hg38 to hg19 coordinates[84], we identified the sites in hg19 that can be uniquely mapped to hg38 and for which the identity of the reference nucleotide does not change after lifting over. Sites not meeting these two specifications were subtracted from the genomic windows of interest.

After applying the filters described above, we were left with the following genes for downstream analyses involving the estimation of mutation density: 14763 protein-coding genes, 3991 divergent lncRNAs (of which 170 are enhancer lncRNAs, 3490 promoter lncRNAs and 331 others), 8454 intergenic lncRNAs (of which 3802 are enhancer lncRNAs, 1177 promoter lncRNAs and 3475 others), 1660 pseudogenes and 1479 genes of other types. Supplementary Data 3 contains the identifiers and FANTOM5 coordinates of all these genes together with their corresponding number of windows and sites per relative position to the TSS or TTS.

**Labelling putative regions under selection.** Exons from GENCODE version 19 and conserved sites as predicted by GERP scores from transcribed and neighbouring untranscribed regions were identified as putatively selected regions[85,86]. We produced versions of the genomic windows where these putatively selected regions were subtracted from the set of analysed sites (windows without conserved elements) or where they were left as the only sites to analyse (windows with only conserved elements). Throughout the manuscript, we note when we have used windows with or without these additional modifications.

### Mutation data sets
**gnomAD and UK Biobank.** Germline variants were collected from gnomAD v3 (https://gnomad.broadinstitute.org/) and from UK Biobank (www.ukbiobank.ac.uk)[21,87]. We used a liftover to convert both gnomAD and UK Biobank data from hg38 to hg19 coordinates. The derived allele frequency spectrum from gnomAD was partitioned into six frequency intervals. Alleles occurring in 1–14 copies (or < 0.01%) were considered extremely rare variants, followed by alleles between 15–140 (0.01–0.1%), 141–1400 (0.1–1%), and 1401–14000 (1–10%). Common polymorphisms were defined as alleles occurring between 14001–126000 (10–90%), and high-frequency polymorphisms were defined as alleles occurring in more than 126000 copies (top 10%). We downsampled the original variable sample size at each site to 140000 chromosomes and discarded sites with a sample size lower than 140000 chromosomes (those represented less than 5% of all segregating sites). For the UK Biobank data, we used approximately 300000 chromosomes and kept only variants with an allele frequency lower than 0.01% (ERVs). There are 414,433,805 gnomAD ERVs, 4,814,941 gnomAD common variants, and 520,004,351 UK Biobank ERVs before restricting to mutations in our genomic windows of interest.

**Cancer data.** Somatic mutations were retrieved from the PCAWG (Pan-Cancer Analysis of Whole Genomes) database[22]. We removed hypermutable tumours of skin, colorectal, and a few other tissue types described in ref. 55. The identifiers of the patients used in this study are listed in Supplementary Data 4. There are 22,587,102 mutations across cancer types before restricting to mutations in our genomic windows of interest.

**De novo and mosaic mutation data sets.** Mutations from parent-offspring or multi-generation sequencing and somatic mosaic mutations from deep sequencing were collected across 13 studies. Mutations shared between studies or between supplementary tables of the same study were filtered such that all data sets contain only unique mutation calls.

A total of 10856 early mosaic variants were collated from

1. Supp. Table 3 of Ju et al. (2017)[43], containing low-VAF mutations from blood (163 unique mutations)
2. the file `gonosomal.dnms` of Sasani et al. (2019)[35] (440 unique mutations)
3. Supp. Table 1 of Jonsson et al. (2021)[33], containing near-constitutional (VAF > 45%) mutations from blood that differed between monozygotic twins, excluding mutations shared by the twins (874 unique mutations)
4. Supp. Table 2 of Jonsson et al. (2021)[33], containing mutations detected in both soma and germline of one monozygotic twin, the latter evidenced by transmission to an offspring (3842 unique mutations)
5. Supp. Table 3 of Jonsson et al. (2021)[33], containing mutations present in both twins with < 25% VAF or significant VAF difference between twins (85 unique mutations)
6. Supp. Table 2 of Rodin et al. (2021)[44], containing predicted non-constitutional SNVs from brain (2166 unique mutations)
7. Supp. Table 3 of Maury et al. (2024)[45], containing SNVs with VAF < 0.4 from brain, presumed to have arisen during prenatal neurogenesis (3286 unique mutations).

We gathered a total of 4764 unique late mosaic variants. These are defined as variants that were reported as de novo variants in the following family sequencing studies and that were shared between siblings (but not present in parental blood):

1. Supp. Table 15 of Goldmann et al. (2016)[37] (26 unique mutations)
2. Supp. Table 3 of Yuen et al. (2017)[38] (2555 unique mutations)
3. Supp. Table 4 of Jonsson et al. (2017)[34] (84 unique mutations after removal of 207 variants shared with Halldorsson et al., 2019)
4. Supp. Table 2 of An et al. (2018)[39] (1257 unique mutations)
5. Goldmann et al. (2018)[36], obtained directly from the authors, but also available at dbGaP accession phs001522.v1.p1 (9 unique mutations)
6. the file post-pgcs.dnms of Sasani et al. (2019)[35] (303 unique mutations)
7. the file aau1043_datas5_revision1.tsv of Halldorsson et al. (2019)[40] (455 unique mutations)
8. Supp. Table 2 of Jonsson et al. (2021)[33] (75 unique mutations).

A total of 774930 de novo variants (DNMs), i.e., variants identified in the following studies that were not shared between siblings, were collated from

1. www.nlgenome.nl of Francioli et al. (2015)[41] (11016 unique mutations)
2. Supp. Table 3 of Goldmann et al. (2016)[37] (35730 unique mutations)
3. Supp. Table 3 of Yuen et al. (2017)[38] (113960 unique mutations)
4. Supp. Table 4 of Jonsson et al. (2017)[34] (9468 unique mutations, after removal of 88721 variants shared with Halldorsson et al., 2019)
5. Supp. Table 2 of An et al. (2018)[39] (231362 unique mutations)
6. the file aau1043_datas5_revision1.tsv of Halldorsson et al. (2019)[40] (180287 unique mutations)
7. Goldmann et al. (2018)[36], obtained directly from the authors, but also available at dbGaP accession phs001522.v1.p1 (73737 unique mutations)
8. the files second_gen.dnms and third_gen.dnms of Sasani et al. (2019)[35] (27684 unique mutations)
9. Supp. Tables 2 and 3 of Richter et al. (2020)[42] (72096 unique mutations, after removal of 88279 variants shared with An et al., 2018)
10. Supp. Table 2 of Jonsson et al. (2021)[33] (19590 unique mutations, after removal of 369 variants shared with Halldorsson et al., 2019).

### Mutation density across genomic windows

**Computation of $\mu$ and $\mu\prime$.** The observable $\mu$ measures the observed mutation probability of a given genomic locus, corrected for the sequence composition of the locus. It is equivalent to the observed number of mutations in a locus divided by the number expected under a uniform distribution. The expected number of mutations is derived from the mutational matrix which represents the global average pentanucleotide-sequence-context dependence of all active mutational processes. That is, the matrix element $m_{ij}$ quantifies the probability of the central base in the pentanucleotide of type $i = 1, ..., 1024$ (corresponding to AAAAA, AAAAC, ... , TTTTT) on the coding (i.e., non-transcribed) strand to mutate to state $j = 1, ..., 4$ (corresponding to A, C, G and T):

$$m_{ij} = \frac{\sum_{l=1}^{L} n_{ij}^l}{a_i}, \tag{1}$$

where $n_{ij}^l$ is the number of mutations of type $(i, j)$ at site $l = 1, ..., L$ and $L$ is the total sequence length. The abundance of the pentanucleotide context $i$ among all $L$ sites is given by $a_i = \sum_{l=1}^{L} I_{c_l}(i)$, with $I_{c_l}(i)$ denoting the indicator function and $c_l$ the pentanucleotide at site $l$,

such that $\sum_{i=1}^{1024} a_i = L$. The summation is taken over all $L$ genomic sites considered in a given analysis. By default, this includes all sites in the (up to) 50 kb untranscribed sequence upstream of the TSS, the (up to) 50 kb transcribed sequence downstream of the TSS, the (up to) 50 kb transcribed sequence upstream of the TTS and the (up to) 50 kb untranscribed sequence downstream of the TTS of each gene. All 4 gene categories (14763 protein-coding, 3991 divergent lncRNAs, 8454 intergenic lncRNAs, and 1660 pseudogenes) were processed together. When a gene was less than 100 kb in length, double-counting of mutations on the transcribed region was avoided by assigning each mutation to whichever element it was closer to out of TSS and TTS. When conserved sequences were excluded from the analysis, the matrix was correspondingly recalculated from only those parts of the sequence that remained. When using a window size of 100 bp (instead of 1 kb), the considered regions were correspondingly reduced to 5 kb (instead of 50 kb). For each data set (UKB ERVs, gnomAD ERVs, gnomAD common variants, DNMs, and pan-cancer), a separate mutation probability matrix was derived.

To compute $\mu_{tb}$ in a given genomic window $b$ of a given gene $t$, we divided the observed mutation count in the window by the expected, as computed from the matrix:

$$\mu_{tb} = \left( \frac{\sum_{l=1}^{W} \sum_{i,j} n_{ij}^l}{\sum_{l=1}^{W} \sum_{i,j} m_{ij} I_{c_l}(i)} \right)_{tb}, \tag{2}$$

where $W \in \{100, 1000\}$ is the size of the window. In other words, all observed mutations in the window were counted and this count was divided by the number of mutations expected in the window based on the genome-wide average mutation probabilities.

We then took the average of $\mu_{tb}$ across all genes to arrive at our observable $\mu_b$ (Supplementary Fig. 9). Since we separated mutations into two categories (non(CpG > TpG) and CpG > TpG), removed conserved sequences (including exons) for some analyses and a gene's TTS may not coincide with the 3' boundary of the window it is located in, we accounted for the increased amount of noise in $\mu_{tb}$ coming from genes that have fewer than $W$ sites in a given window by weighting each value with the fraction of sites considered coming from the gene, $w_{tb}$:

$$\mu_b = \frac{\sum_t \mu_{tb} w_{tb}}{\sum_t w_{tb}}. \tag{3}$$

To increase the information content of the figures, we also derived a second observable, $\mu\prime$. Beyond the pentanucleotide sequence context, $\mu\prime$ also corrects for the regional background mutation probability of the genomic locus in which the gene is embedded since the latter correlates with gene length and regional gene density. To this end, for each window in a given gene $t$, we divide that window's $\mu_{tb}$ by the ratio $g_t$ of the observed and expected mutation count of the entire gene,

$$g_t = \left( \frac{\sum_{l=1}^{\ell_t} \sum_{i,j} n_{ij}^l}{\sum_{l=1}^{\ell_t} \sum_{i,j} m_{ij} I_{c_l}(i)} \right)_t, \tag{4}$$

where $\ell_t$ denotes the total length of regions considered from the given gene, including upstream TSS untranscribed, downstream TSS transcribed, upstream TTS transcribed, and downstream TTS untranscribed.

**Error Bars for $\mu$ and $\mu\prime$.** To estimate the sampling variance around our estimates of $\mu$ and $\mu\prime$, we derived simple percentile confidence intervals from bootstrap samples[88]. Briefly, for a given gene type, we sampled with replacement as many genes as analysed together with their corresponding observed mutations and repeated this 100 times. $\mu$ and $\mu\prime$ were calculated after each resampling iteration as described in the

previous section. We found the values representing the percentile 5 and 95 of the resulting bootstrap distributions for each window and defined the 90% confidence interval as the range of values between them. Error bars in Fig. 1 and in similar plots represent these intervals and points are the means of the bootstrap distributions.

**Mutation density estimates for mosaic variants.** To study the mosaic mutation density at different genomic windows, we introduced a variant of the analysis consisting of estimating a mutational matrix with only 12 mutation types representing the possible mononucleotide substitutions without considering sequence contexts across sites. This matrix is used in place of the pentanucleotide context matrix described in previous sections to compute the expected number of mutations per window. We do this in the context of Fig. 2f due to the low number of mutations in mosaic data sets which would result in an extremely sparse pentanucleotide matrix.

The `glm` function from the R `stats` package with `family = Poisson` and all other parameters left as defaults was used to automate the estimation of mutation density and significance testing via the `summary` method[89]. An interceptless model was fit with the mutation count as the response variable, the window position relative to the TSS as the only explanatory variable, and the logarithm of the expected mutations as an offset. The resulting coefficient for each set of windows is equivalent to the logarithm of the sum of observed mutations across the windows of the set divided by the corresponding sum of expected mutations. A 2-sided z-test with the null hypothesis that the model's coefficients are equal to 0 was performed (p-values reported in the main text). 95% confidence intervals were derived assuming normality of the sampling distribution of the coefficients using the `confint` method. Coefficients and interval limits were exponentiated back to the original scale of the observed-to-expected fold change of mutations for interpretability purposes in Fig. 2.

**Signature decomposition analyses**
To identify genomic window-specific candidate mutational processes contributing to the observed mutations, we ran a signature decomposition analysis using non-negative least squares (NNLS) optimisation as implemented in the `nnls` R package[90,91]. Pooled mutation counts across genes were calculated according to the positions of the windows of origin of each mutation relative to the TSS. We leveraged two different sets of signatures: COSMIC v3.4[59] and the germline components from Seplyarskiy et al.[24].

COSMIC mutational signatures were extracted using whole-genome (WG) cancer data and were not normalised by the trinucleotide abundance of the genome. Since our analysis is restricted to genomic windows, we need to re-scale our data to account for their different trinucleotide composition. To do so, we divided each trinucleotide mutation type by its window-specific abundance and multiplied it by the WG abundances. This re-scaled input is the one we used for the NNLS decomposition. Signatures with an assigned weight lower than a cutoff (between 0.01 and 0.1, depending on the study and the bin) were not plotted, and their weights were aggregated into a category called Others.

Since genomic windows with very few mutations cannot be reliably decomposed using NNLS, we restrict the decomposition to windows with at least 1000 mutations. Moreover, in order to account for the effect of the number of mutations on the quality of the decomposition when comparing different windows, we downsample the number of mutations in each window to the number of mutations in the window with the lowest count. Finally, to compare the mutation load associated with the signatures across windows, we multiply the signature weights of each window by the total number of mutations in the window and divide by the total number of analysed sites in the window, giving the mutation density.

The germline components were extracted from extremely rare variants[24]. These components were normalised by the trinucleotide abundance and are available in two different formats: extracted from data normalised by the standard deviation and without normalising. We used the non-normalised version for our analyses. Moreover, these signatures have 192 mutation categories, compared to the standard 96 used in the COSMIC signatures. This is because mutation types were not collapsed to the pyrimidine bases in order to analyse strand biases. Therefore, when decomposing samples with these components, we do not collapse the mutation types and we normalise the input by the trinucleotide abundance of the windows. To compare the component weights in ERVs and DNMs, we pooled mutations from 20 kb around the TSS (10 kb to either side) and decomposed all these mutations together, for each data set separately.

**Genomic features**
Listed below are the 38 genomic features used throughout this work together with their sources. We aimed to assign one value or label per feature to each of the genomic windows of up to 1 kb (only with the transcription filter defined in the Transcription, Mappability and Liftover Filters section). When required, we used the Big Binary Indexed suite of tools to convert files to BED format[92]. Other tools used for data parsing and genomic coordinate intersection were `wig2bed` and `bedtools`[93,94]. When necessary, we lifted genomic coordinates from hg38 to hg19 using `liftover`[84]. Most of the source files we use can be accessed through the University of California Santa Cruz (UCSC) genome browser or Gene Expression Omnibus (GEO) resources[95,96]. Unless otherwise specified, for non-categorical features based on various signal intensities, we intersected the reported regions in the source files with our genomic windows and computed the per-base signal density in each window assuming a signal of 0 when a region was not reported in the files. Features denoting properties of genes had their values or labels propagated to all windows associated with a given gene.

- Huvec and H1hesc histone marks: we averaged the signal files across both cell lines of 11 different histone marks measured by the ENCODE project[97]. These include H3K9me3, H3K4me1, H3K4me2, H3K4me3, H3K9ac, H3K9me1, H3K27ac, H3K27me3, H3K36me3, H3K79me2, and H4K20me1. Signal file contents and download locations are documented at http://genome.ucsc.edu/cgi-bin/hgFileUi?db=hg19&g=wgEncodeBroadHistone.
- CpG methylation: high-coverage methylome from the iMETHYL database (http://imethyl.iwate-megabank.org/downloads.html)[98]. The methylome of human blood cells (monocytes, CD4T, and neutrophils) was downloaded from the iMETHYL database and used as a proxy for the methylation status of CpG dinucleotides in the human genome across tissue types.
- DNase I Hypersensitivity Sites (DHS): DHS files were downloaded for the tissue of origin of each tumour type. As a proxy for germline DHS, we use the H1 human embryonic stem cell line. Data originate from the ENCODE project and are deposited as the file `wgEncodeRegDnaseClusteredV3.bed.gz` at http://hgdownload.soe.ucsc.edu/goldenPath/hg19/encodeDCC/wgEncodeRegDnaseClustered/?C=M;O=D[97]. To calculate the pan-cancer DHS signal, we took the weighted mean signal across tissues of origin according to the number of tumours from each tissue type in PCAWG (see Supplementary Data 5).
- Genomic Evolutionary Rate Profiling (GERP) score: conserved elements according to the GERP score were retrieved from the Sidow Lab at Stanford University (http://mendel.stanford.edu/sidowlab/downloads/gerp/index.html)[86].
- Somatic double-strand breaks (DSB): we obtained a high-sensitivity single-nucleotide resolution somatic DSB map for NHEK cells from the work of Lensing et al (https://www.ncbi.nlm.

nih.gov/geo/query/acc.cgi?acc=GSE78172, file `BREAK_primar-y_n1.NOdups.nomdel. default_peaks.narrowPeak.gz`)[49].

- Huvec cells H2A.Z histone mark: we downloaded the ENCODE signal file documented at https://genome.ucsc.edu/cgi-bin/hgTables?db=hg19&hgta_group=regulation&hgta_track=wgEncodeBroadHistone&hgta_table=wgEncodeBroadHistoneHuvecH2azSig&hgta_doSchema=describe+table+schema[97].

- Nucleosome density: we retrieved high-resolution lymphoblastoid B-cell MNase-seq stable nucleosome regions from Gaffney et al.[99]. We focused on the normalised nucleosome occupancy signal provided by NucPosDB (https://generegulation.org/NGS/stable_nucs/hg19/, file `GSE36979_Gaffney2012_Bcells_MNase-seq_stable_100bp_hg19.bed.gz`)[100].

- Meiotic recombination hotspots associated with PRDM9: we retrieved a signal of human testis meiotic double-strand break hotspots found through their relationship with the binding of DMC1 at PRDM9 binding motifs from Supp. Table 1 of Hinch et al.[46]. We extended the coordinates of the hotspots 100 bp in both the 5' and 3' directions to capture the main peak of the mutation footprint around hotspots that is reported by Hinch et al.[46]. The signal intensity reported in the data was used for all 201 bp associated with each hotspot after extending the annotations.

- Replication time: we computed the mean wavelet-smoothed signal across 6 cell lines measured by the ENCODE project[97]. The cell lines are: Gm12878, Helas3, Hepg2, Huvec, K562, and Mcf7. The data files and download locations are documented at https://genome.ucsc.edu/cgi-bin/hgTrackUi?db=hg19&g=wgEncodeUwRepliSeq. We intersected the reported regions with our genomic windows and computed the per-base signal density in each window. If a base had no reported replication time, the base was ignored altogether during the calculation.

- Recombination rate: the average genetic map of recombination rates computed from the paternal and maternal maps was retrieved from the results of Kong et al. documented at https://genome.ucsc.edu/cgi-bin/hgTables?db=hg19&hgta_group=map&hgta_track=decodeRmap&hgta_table=decodeSexAveraged&hgta_doSchema=describe+table+schema[101]. The per-base density of this feature was calculated for each genomic window similarly to that of replication time.

- Replication strand: we took the data from the Mcf7 cell line used for calculating the replication time feature and leveraged it to compute a replication direction map using the `reptDir` R package[102]. The method used by the package is related to the descriptions provided by previous work on replication direction[8,103], and is documented in the cited package. We specified a minimum replication direction domain length of 250 kb and a minimum slope of 0.01 between individual regions in the domains. Then, we used the obtained direction map to determine if each genomic window was replicated in a single direction, in a mix of both, or if it had base pairs for which the direction was unreliable. For windows replicated in a single direction, we were able to determine the identities of the leading and lagging strands of replication. As mutations were counted in the coding strand of transcription, we created a categorical variable with 3 possible labels indicating if the coding strand is also the leading strand, the lagging strand, or if such thing is unknown as is the case for windows with base pairs replicated in an unreliable direction or with a mix of base pairs replicated in both directions.

- G/C skew: we calculated the difference between the number of guanines and cytosines for each genomic window and divided it by the sum of both nucleotide counts. The computation is applied to the template strand of transcription according to the annotations of the particular gene associated with each window.

- CpG content: derived from the number of CpG sites in the genome (https://doi.org/10.6084/m9.figshare.1415416.v1)[104]. This is the count of sites that are found in each genomic window divided over the total bases in the window.

- G/C content: this is the simple sum of the number of guanines and cytosines in each genomic window divided by the number of total bases in the window.

- CpG island (CpGI) content: we defined CpGIs using the annotations file available at http://genome.ucsc.edu/cgi-bin/hgTrackUi?hgsid=1102630601_okTYJ1lN4a4kGQ6CPBEtw3g218bg&c=chr1&g=cpgIslandExtUnmasked[105]. We intersected the coordinates with our genomic windows and computed the fraction of CpGI bases in each window.

- Gene with CpGI promoter: promoters of each gene were defined as regions ranging from −1.2 kb to +0.3 kb relative to the TSS of BioMart Ensembl75 annotations[106]. The CpGI annotations cited for the previous feature were intersected with these promoters. A binary categorical variable was defined depending on whether or not at least 1 bp of the promoter intersected with a CpGI.

- Density of G-quadruplexes: genomic coordinates of G-quadruplexes in the human genome with experimental evidence were downloaded from the EndoQuad database (http://chenzxlab.hzau.edu.cn/EndoQuad/#/download, file `Human_eG4.txt`)[107]. We computed the number of bases that intersected each genomic window and divided it by the total number of bases in the window.

- Gene TATA box score: we retrieved the position frequency matrix for the TATA box motif from the JASPAR database (https://jaspar.elixir.no/matrix/MA0108.3/)[108]. Each matrix element was divided by the sum of elements in the column where it is located. The result was further divided by 0.25 which represents the background frequency for all 4 nucleotide types. Then, to obtain the Position Weight Matrix (PWM)[109], a pseudocount of 1 was added to all elements before taking their base 2 logarithms. We obtained the sequences in the genome found in sliding windows of the same size as the TATA box motif between -40 bp and -20 bp of the TSS of each gene[110]. The PWN was used to score each sliding window and their reverse complements. Finally, the TATA box score is the maximum score across all sliding windows of each gene.

- Gene with upstream divergent promoter: paired TSS annotations derived from Global Run-on cap (GRO-cap) sequencing data obtained from k562 cells were retrieved from the Supp. Data set of Core et al.[52]. Each pair annotates a GRO-cap TSS in the plus strand that is located 150 bp or less from another GRO-cap TSS in the minus strand. We defined the putative divergent promoter region as the gap between the two GRO-cap TSS defined in each pair. In a minority of cases, the coordinates of the two GRO-cap TSS overlapped and we took the putative divergent promoter as the overlapping region. We then defined a binary categorical variable depending on whether or not putative divergent promoters overlapped with the first genomic window upstream of the TSS belonging to each gene.

- Gene with downstream divergent promoter: this feature is similar to the one defined above with the only difference being that putative divergent promoters are intersected with the first bin downstream of the TSS.

- Gene length: corresponds to the total length of the whole gene of origin of each genomic window. The lengths include intronic regions and are computed directly from the FANTOM gene models[80].

- Gene expression: as a proxy of gene expression in the human germline, and given that more than 75% of DNMs are from paternal origin[111], we used single-cell transcriptomic unique molecular identifier counts from male germline cells provided upon request by Xia et al.[9]. For somatic expression, we considered data from the

Genotype-Tissue Expression (GTEx) project version 4 (https://dcc.icgc.org/releases/PCAWG/transcriptome/transcript_expression/, file `GTEX_v4.pcawg.transcripts.tpm.tsv.gz`)[112,113]. We averaged the expression of each gene across samples of the same tissue using the metadata file `GTEX_v4.metadata.tsv.gz` available at https://dcc.icgc.org/releases/PCAWG/transcriptome/metadata[113]. Supplementary Data 6 shows the number of samples averaged for each tissue type and the correspondence with PCAWG cancer types. To obtain the pan-cancer expression, we applied the same procedure as described for the DHS feature using the mean expression across tissues as input for the weighted average.

- Gene is essential gene: essential genes found by Wang et al. and processed according to Weghorn and Sunyaev were retrieved from the supplementary files of the latter authors[48,114]. We defined a binary categorical variable denoting whether or not a gene is in this set of essential genes.
- Gene is Tumour Suppressor Gene (TSG): TSGs without a dual role as oncogenes were extracted from the Cancer Gene Census (CGC) previously available at https://cancer.sanger.ac.uk/census[115] (560 genes total before filtering). Both tiers 1 and 2 were considered. We defined a binary categorical variable denoting whether or not a gene is in this set of TSGs.
- Gene is oncogene: similar to the previous feature but for oncogenes without a dual role as TSGs.
- Gene Precision Run-on Sequencing (PRO-seq) ratio: we obtained strand-specific maps of single-nucleotide resolution engagement of RNA polymerase II complexes in K562 cells from the file `GSE60456_RAW.tar` at https://www.ncbi.nlm.nih.gov/geo/query/acc.cgi?acc=GSE60456[116]. The reads of these PRO-seq data originating from the transcribed strand of each gene were identified. We summed the reads counted in the first 500 bp of the first genomic window downstream of the TSS of each gene. We did the same for the reads counted in the remainder of the aforementioned window and the rest of the downstream TSS windows. We then divided these sums by the appropriate sum of sites to obtain the per-site PRO-seq signal density at the beginning and at the gene body of each gene. We added a pseudocount of 1 to these values before computing the difference between the base 10 logarithms of the beginning and gene body signals. The result is the final PRO-seq ratio feature for each gene.
- Gene from maternal, non-maternal, or other chromosomes: a categorical variable with three possible labels denoting the chromosome group where the gene is found was specified. The definition of the groups was guided by the results of Seplyarskiy et al. who showcase a mutational process acting on the human germline that is enriched with clustered mutations of maternal origin[24]. We defined the chromosome groups according to the fraction of maternal mutation clusters per chromosome that are shown in the cited work. We focused on the four most extreme chromosomes on both sides of the spectrum. Groups consist of maternal chromosomes (8, 9, 15, and 16), non-maternal chromosomes (1, 13, 18, and 20), and other chromosomes.

### Regression analyses
**Modelling of mutation density.** We model the mutation density across genomic windows in the data sets of interest as a variable dependent on predictors derived from genomic features. This is done within a Generalised Linear Model (GLM) framework using a log link function[117]:

$$\ln(\mathbf{y}) \sim \boldsymbol{\beta}\mathbf{X} + \ln(\mathbf{u}) \tag{5}$$

where **y** is a $n$-dimensional vector containing mutation counts for the different genomic windows (the numerator of Equation (2)). **X** is a $n$-

by-$p$-dimensional design matrix, $\boldsymbol{\beta}$ is a $p$-dimensional vector of regression coefficients which are to be estimated from the data and **u** is a $n$-dimensional vector of exposures (denominator of Equation (2)). The $\ln(\mathbf{u})$ term acts as an offset that adjusts the mutation counts exactly by the expected number of mutations in the corresponding window[117].

Under the assumption that the conditional errors around the model's predictions are Poisson distributed, we estimated the regression coefficients using the `glm` function from the `stats` R package with `family = poisson` or using the `glm.nb` function from the `MASS` R package (Poisson and negative binomial regression, respectively)[89,118]. Negative binomial regression is useful in the presence of over-dispersion as it introduces additional variance by allowing extra gamma-distributed errors dependent on a single additional parameter that can be estimated together with the regression coefficients[118]. For all non(CpG > TpG) mutation data sets, negative binomial regression was used. For CpG > TpG mutation data sets, attempts to fit a negative binomial regression model tended to not converge due to a lack of overdispersion in these cases[119]. Therefore, Poisson regression was applied to these data sets. Parameters for the estimation routines were left as defaults except for the number of maximum iterations, which was increased to 1000.

**Design matrix and regression coefficient interpretation.** In the context of the regression results shown in Fig. 3, we aimed to describe changes in mutation density across genomic windows as a function of both position-unspecific changes (main effects) and position-specific changes in genomic features (interaction effects)[120]. To achieve this, the design matrix was defined with matrix augmentation as:

$$X = [\mathbf{1}^T \, \mathbf{A} \, \mathbf{B} \, \mathbf{C}] \tag{6}$$

Where $\mathbf{1}^T$ is a column-vector of ones. **A** is a matrix composed of $m = 1, \ldots, M$ ($M$ = number of features) column-vectors containing the genomic feature information of all windows such that $\mathbf{A} = [x_1 x_2 \ldots x_m]$. **B** is a matrix of $k = 1, \ldots, K$ ($K$ = number of position categories) column-vectors encoding the position category membership of any particular window such that $\mathbf{B} = [b_1 b_2 \ldots b_k]$, and **C** is a matrix containing all Hadamard products between pairwise combinations of column-vectors in **A** and **B** such that $\mathbf{C} = [(x_1 \odot b_1)(x_1 \odot b_2) \ldots (x_m \odot b_k)]$. The number of column-vectors in **X** is therefore $p = 1 + m + k + mk$.

Categorical variables were included in **A** through dummy coding and position categories in **B** through sum coding[121]. Overall, this design allows us to make the following interpretations:

- Main effects: they are represented by the regression coefficients $\hat{\beta}_1, \hat{\beta}_2, \ldots, \hat{\beta}_m$. These are the average changes in the log mutation density across position categories per unit change in a non-categorical genomic feature. For categorical features, the coefficient represents a similar change when switching between the reference and non-reference feature labels. For almost all of these features, the reference is no and the only coefficient associated with each feature reflects the change to yes. The only exceptions are maternal chromosome and replication strand status, where there are two possible non-reference labels and hence two coefficients for each of these features instead of one. The reference labels for these features are other and unknown, respectively (see the Genomic Features section).
- Interaction effects: they are comprised by regression coefficients $\hat{\beta}_{m+k+1}, \hat{\beta}_{m+k+2}, \ldots, \hat{\beta}_p$. These are additional changes in log mutation density on top of the main effects that also depend on the set of features in **A** but only apply in the context of specific position categories.

In typical parametric regression fashion, the interpretation of coefficients associated with a particular genomic feature is done

considering all other features that have been adjusted for (i.e., held at some constant value). Note that by taking any coefficient and computing $100 \times (e^\beta - 1)$, one can obtain the corresponding percentage change in the original unlogged mutation density scale.

For regressions in Fig. 6 and Supplementary Fig. 13, we used a similar approach as described above but the **C** matrix component of the design matrix was omitted (i.e., models with no interactions), and the matrix **B** was instead reworked into a simple integer vector denoting distance from the TSS (predictor denoted as bin). For regressions in Fig. 3b, including all genomic features in **X** causes convergence problems while fitting the models, due to the low number of mutations in some of the data sets considered in this analysis. Therefore, in this case, we opted to fit one regression per feature keeping the design with main and interaction effects.

To simplify the coefficient interpretation of the numerous non-categorical features that we include in each regression, each with different units of measurement, standardised regression coefficients are provided for all analyses[122]. To achieve this, we simply mean-centred and variance-scaled the non-categorical predictor column vectors in **A** after filtering out windows with missing data for any of the features and before calculating **C** when this matrix was needed.

**Addressing multicollinearity and model diagnosis.** For all regressions that considered more than one genomic feature at a time, we performed two procedures to rule out major problems confounding the estimation of regression coefficients and their associated inference procedures:

- Multicollinearity is associated with unstable estimation of coefficients and inflated standard errors that can obscure model interpretation and inference[123]. To prevent strong multicollinearity, we applied the following procedure: with the whole set of 38 features, a simplified model with no interactions was fitted (**C** in Equation (6) omitted from the design matrix). Then, the Generalised Variance Inflation Factor (GVIF) was calculated for each feature using the `car` R package[124]. To make non-categorical and categorical features comparable, we took the GVIF to the power of the reciprocal of the degrees of freedom associated with each feature as the multicollinearity estimator[125]. The feature with the maximum value of this estimator was identified and if its value exceeded a predefined threshold, it was removed and a new reduced model was fitted. We repeated this procedure iteratively until no features exceeded the threshold. With the set of remaining features, we fitted the final models for each data set including at this point the **C** matrix in the design if necessary. We worked with a threshold of 5 as a middle ground between conservative values and more lax criteria[126].

- Model misspecification can manifest as severe overdispersion or underdispersion of the data compared to the model's predictions[127]. Particularly, overdispersion leads to underestimation of coefficient standard errors and therefore false positives when performing inferences[128]. After fitting each regression, we examined the residual dispersion statistic provided by the `DHARMa` R package[129]. We found that in most regressions, we observe only slight deviations from equidispersion. In the cases where stronger deviations exist, they trend towards underdispersion which has the effect of making the inference more conservative[127].

In Supplementary Data 1–2, we provide the list of genomic features that were filtered out due to the GVIF threshold for each regression shown. We also provide the dispersion statistics along with other summary statistics such as the number of mutations and number of windows involved. For all reported regression results, we ensured there were no warnings or errors raised by the algorithms estimating the regression coefficients.

**Regression coefficient hypothesis testing and inference.** The aim of the regression analyses presented in Fig. 3 is to compare the position-specific effects of genomic features on mutation density within 1 kb from the TSS and at distances 6 kb or farther from the TSS. As a result, vectors of matrix **B** in Equation (6) encode window membership to four position categories: downstream TSS, downstream far, upstream TSS, and upstream far. This entails that each predictor derived from the genomic features is associated with four position category interactions.

After estimation of the regression coefficients, for each predictor, we computed the difference between TSS and far interaction effects associated with the predictor for downstream and upstream positions. This corresponds to comparing the effect on log mutation density that is exclusively attributable to changing the predictor at the TSS against the analogous effect far from the TSS. To complement these statistics, we also calculated the total effect of changing each predictor at the TSS by summing the corresponding interaction effect with the main effect of the predictor. For assessing the significance of these linear combinations of regression coefficients, we assumed normality of the sampling distribution of the estimates to leverage the `multcomp` R package[130]. The package's routines allow for computing $p$-values based on z-tests under the null hypothesis that arbitrary sums or differences of coefficients equal 0 while taking into account the covariance structure between coefficients. Furthermore, $p$-values and confidence intervals can be adjusted for multiple testing to control the family-wise error rate within each regression[130]. We performed the adjustment using the default parameters of the `summary` and `confint` methods for `glht` objects except for `test = adjusted(maxpts = 10 * 50000)`, which was needed for successful calculation of $p$-values when testing many simultaneous hypotheses.

For regressions in Fig. 6 and Supplementary Fig. 13, we focused on all 1-kb windows found within 50 kb upstream or downstream from the TSS. For each mutation data set, we fitted one separate regression for each of upstream and downstream windows without interactions. To assess the multiple testing-controlled significance of each predictor, we applied the same approach described above to test the null hypotheses that individual regression coefficients are equal to 0.

## Mutation density strand bias across genomic windows

We defined different mutation categories based on the type of mononucleotide substitution (A > G, T > G, A > T, G > T, C > G and C > T), on the strand where the mutation is counted relative to transcription (coding and template), and based on the replication strand annotations described in the Genomic Features section. Mutations of each type were summed across all genes according to the genomic window in which they appear. For each window, the resulting mutation counts were divided by the corresponding sum of the number of nucleotides that can lead to each type of mutation. The resulting mutation densities are grouped by mononucleotide substitution type and displayed in panels of Fig. 4e (gnomAD data), Supplementary Fig. 17 (PCAWG data) and Supplementary Fig. 18 (gnomAD data, 100 bp).

## Evolution analyses

**dN/dS ratio across genomic windows.** In order to have a reading frame of reference for each gene, we used coding regions from GENCODE version 19 whose length is a multiple of 3 to determine the effect of mutations on amino acids[85]. For simplicity, the transcript used for each gene was simply the one specified in the gene expression data we used (see the Genomic Features section). Borrowing from the definitions of Equation (2), we denote the number of mutations of type $x$ in transcript $t$ and genomic window $b$ as $d_{tb}^x = \left( \sum_{l=1}^{W} \sum_{i,j} n_{ij}^l \delta_x(l,j) \right)_{tb}$ where $\delta_x(l,j)$ is a function that evaluates to 1 when the mutation to state $j$ at site $l$ produces a variant of type $x$ in transcript $t$ and to 0 otherwise. The two types of possible mutations $x \in \{z, s\}$ are non-synonymous and

synonymous, respectively. Similarly, we define the expected number of mutations of each type as $h_{tb}^{x} = \left( \sum_{l=1}^{W} \sum_{i,j} m_{ij} I_{c_l}(i) \delta_x(l,j) \right)_{tb}$.

To obtain the dN/dS ratio for window $b$ we evaluate the function:

$$r(b) = \frac{\sum_t d_{tb}^z / \sum_t h_{tb}^z}{\sum_t d_{tb}^s / \sum_t h_{tb}^s} \tag{7}$$

Potential sampling variance around the ratios was addressed through resampling. Briefly, for each transcript and window, we randomly sampled with replacement as many mutations as observed 100 times. For every resampling iteration, we calculated the dN/dS ratios as described above for each window. Similarly to what was described in the Error Bars for $\mu$ and $\mu\prime$ section, resamples were used to calculate dN/dS confidence intervals and data points shown in Supplementary Fig. 19a (gnomAD data). For DNMs, due to low numbers of synonymous mutations in some windows, sometimes resampling of mutations would yield 0 synonymous mutations. In these cases, the dN/dS ratio cannot be computed. In windows where this happens, we plot only the observed dN/dS ratio estimate without error bars.

**Relative comparison of mean mutation density across genes for ERVs and common variants at different genomic windows.** We computed $\mu_{tb}$ as defined in Equation (2) for the gnomAD ERVs and common variants. Then, we calculated a simple average of $\mu_{tb}$ across genes for each genomic bin to obtain $\mu_{ERV}$ and $\mu_{common}$, respectively. At each bin position, the ratio $\mu_{common}/\mu_{ERV}$ was defined as a statistic to measure the relative change in average mutation density across genes due to the action of selection. Possible sampling variance of this statistic is addressed through resampling. We sampled with replacement as many mutations as observed across all genes and windows for each data set 100 times. We calculated $\mu_{common}/\mu_{ERV}$ as explained above for each window in each resampling iteration. Similarly to what was described in the Error Bars for $\mu$ and $\mu\prime$ section, resamples were used to calculate $\mu_{common}/\mu_{ERV}$ confidence intervals and data points shown in Supplementary Fig. 19b (gnomAD data).

**Effect of GC-biased gene conversion on mutation density across genomic windows.** For each gene and genomic window, we calculated a value similar to $\mu_{tb}$ from Equation (2) but that only takes into account subsets of mutation types. We defined three subsets according to the expected action of GC-biased Gene Conversion (BGC) on mutations: A < > T/C < > G substitutions (BGC-indifferent), AT > GC substitutions (BGC-favoured), and GC > AT (BGC-unfavoured). We continued to calculate the analogous version of $\mu_b$ from Equation (3) for each of the mutation subsets keeping weights proportional to the total number of sites in each window. Then, we also obtained the similarly modified version of $g_t$ from Equation (4) in order to compute the mutation subset-specific $\mu\prime$ of each of the three aforementioned mutation categories. Error bars and data points for these statistics are plotted in Fig. 5b (gnomAD data) and were derived in the same way as described in the Error Bars for $\mu$ and $\mu\prime$ section.

## Gene set analyses

**TSS mutational enrichment of disease phenotype genes.** We found the intersection between protein-coding genes in our study and genes in the Human Phenotype Ontology (HPO) version 2.0.6 terms available in the file phenotype_to_genes.txt (https://hpo.jax.org/data/annotations)[69]. For the intersecting genes with each term, non(CpG > TpG) gnomAD ERVs across the first 1000 bp genomic windows upstream and downstream of the TSS were pooled. The corresponding expected number of mutations described in the Mutation Density across Genomic Window section was similarly pooled. The overall density of TSS mutations per HPO term was calculated as the division of the pooled observed over expected values.

By applying a similar procedure to all other 1000 bp windows in our study, including TTS-anchored windows and TSS-anchored windows other than the ones closest to the TSS, we obtained the mutation density in non-TSS regions per HPO term. We limited our analysis to HPO terms with 100 or fewer genes to be more specific about the examined phenotypes (Supplementary Data 7).

Figure 5c colours data points by the TSS to non-TSS ratio of mutation densities. These densities can be understood as Poisson rate estimates since they give an expectation for the number of mutations that should be observed per mutation chance. We use an exact conditional binomial test to check if the ratio of these rates is significantly different from 1[131]. To be conservative about our inferences and correct for testing multiplicity, the resulting two-sided $p$-values for HPO terms are adjusted using the Bonferroni correction[132]. The statistical test and $p$-value correction are both implemented in the stats R package as the poisson.test and p.adjust functions, respectively[89].

**Overrepresentation of genes with TSS mutational enrichment across biological pathways.** We pooled the observed and expected number of mutations for TSS and non-TSS genomic windows using the same data and approach as described in the TSS Mutational Enrichment of Disease Phenotype Genes section. However, we did not pool across genes this time but instead calculated the TSS to non-TSS ratio of mutation densities and its $p$-value for every gene separately. We corrected these $p$-values using the Benjamini-Hochberg method implemented in the p.adjust function in the stats R package (argument method = "fdr")[89,133]. Genes with adjusted $p$-value $q < 0.01$ were considered for downstream testing.

We used the EnrichR R package to test if the genes with significant enrichment of TSS mutations relative to mutations elsewhere in our regions of interest were overrepresented in terms from the Gene Ontology Biological Process 2025 (GOBP) collection available in the EnrichR tool[10,134]. Note that we were able to compute the TSS to non-TSS ratio of mutation densities only for genes where such a statistic was properly defined. Therefore, we set the background of genes in the enrichment analysis to those with mutation chances both at the TSS and elsewhere as well as a positive count of mutations at non-TSS regions. We report the two-sided $p$-values adjusted for multiple testing computed by EnrichR to assess the overrepresentation of our input genes in the GOBP gene sets (Supplementary Data 8).

## Correlation between developmental gene expression and TSS mutational enrichment

We obtained human early development single-cell RNA-seq data from Supp. Table 1 of Yang et al. (2013)[135]. We took the average Reads Per Kilobase per Million mapped reads (RPKM) across replicates already provided by the authors. Variance stabilization was performed by taking the logarithm base 10 of these values after adding a pseudocount of 1. The similarity between the gene expression of the different stages was assessed by computing all pairwise Pearson correlations between them while considering all genes available. The corrplot R package was used for ease of visualisation[136].

For each stage, we also computed the Pearson correlation coefficient between the gene expression of each gene and its corresponding gnomAD ERVs mutational enrichment in the TSS relative to other analysed regions. The former was computed exactly as described in the Overrepresentation of Genes with TSS mutational Enrichment across Biological Pathways section. Protein-coding genes with non-missing gene expression and computable TSS mutational enrichment were used. The correlation coefficient of the morulae stage was compared against all others using two-sided Hittner tests, which take into account the correlation between expression profiles of the pairs of stages when assessing the null hypothesis that the difference between correlation coefficients is 0[137]. The implementation of the test in the cocor R package was used[138].

All correlation statistics described in this section are shown in Supplementary Fig. 15. All p-values, including those from t-tests to asses the significance of individual correlation coefficients, were multiple-testing-corrected using the `p.adjust` function in the `statsR` package (argument `method = "bonferroni"`)[89,132].

## Reporting summary

Further information on research design is available in the Nature Portfolio Reporting Summary linked to this article.

## Data availability

All data are publicly available unless otherwise stated. Germline variants were collected from gnomAD v3 (https://gnomad.broadinstitute.org/downloads#v3) and the UK Biobank (www.ukbiobank.ac.uk)[21,87]. UKBB data are restricted and were accessed under application ID 81963. Researchers can apply for UKBB access via the UKBB Access Management System (https://ams.ukbiobank.ac.uk/ams/). Pan-cancer somatic mutations and somatic gene expression were retrieved from the PCAWG (Pan-Cancer Analysis of Whole Genomes) database (https://docs.icgc-argo.org/docs/data-access/icgc-25k-data)[22]. Gene coordinates were downloaded from FANTOM[81]. HPO disease annotations were obtained from https://hpo.jax.org/data/annotations[69]. GENCODE version 19 annotations were fetched from https://www.gencodegenes.org/human/release_19.html[85].

The mappability filter, Huvec and H1hesc histone mark signals, somatic and germline DHS signals, replication time signals, recombination rates and CpG island annotations were fetched from the UCSC genome browser (https://hgdownload.soe.ucsc.edu/downloads.html)[95]. The DSB signals and PRO-seq read counts were downloaded from GEO with accessions GSE78172 and GSE60456, respectively[96]. GERP scores were downloaded from http://mendel.stanford.edu/sidowlab/downloads/gerp/index.html[86]. Somatic mutational signatures were downloaded from COSMIC (https://cancer.sanger.ac.uk/signatures/downloads/) and germline signatures were obtained from the supplementary materials of Seplyarskiy et al. (https://doi.org/10.1126/science.aba7408)[24,59]. Methylation data was obtained from the iMETHYL database (http://imethyl.iwate-megabank.org/downloads.html)[98]. The nucleosome density signal was retrieved from NucPosDB (https://generegulation.org/NGS/stable_nucs/hg19/)[100]. Meiotic recombination hotspots were obtained from the supplementary materials of Hinch et al. (https://doi.org/10.1126/science.adh2531)[46]. CpG Content was downloaded from https://doi.org/10.6084/m9.figshare.1415416.v1[104]. G-quadruplex annotations were fetched from EndoQuad (http://chenzxlab.hzau.edu.cn/EndoQuad/#/download)[107]. The position frequency matrix for the TATA box motif was downloaded from JASPAR[108]. GRO-cap data was obtained from the supplementary materials of Core et al. (https://doi.org/10.1038/ng.3142)[52]. Germline expression was obtained from Xia et al. upon request[9]. Essential genes were obtained from the supplementary materials of Weghorn and Sunyaev (https://doi.org/10.1038/ng.3987)[48]. Oncogene and TSG annotations were obtained from the CGC (https://cancer.sanger.ac.uk/census)[115].

Early mosaic variants, late mosaic variants and DNMs were collated from the supplementary materials of Sasani et al. (https://doi.org/10.7554/eLife.46922)[35] and Jonsson et al. (2021, https://doi.org/10.1038/s41588-020-00755-1)[33]. More late mosaic variants and DNMs were collated from Goldmann et al. (2018, obtained by direct request to the authors)[36] and from the supplementary materials of Goldmann et al. (2016, https://doi.org/10.1038/ng.3597)[37], Halldorsson et al. (https://doi.org/10.1126/science.aau1043)[40], Jonsson et al. (2017, https://doi.org/10.1038/nature24018)[34], Yuen et al. (https://doi.org/10.1038/nn.4524)[38], and An et al. (https://doi.org/10.1126/science.aat6576)[39]. More early mosaic variants were collated from the supplementary materials of Ju et al. (https://doi.org/10.1038/nature21703)[43], Rodin et al. (https://doi.org/10.1038/s41593-020-

00765-6)[44], and Maury et al. (https://doi.org/10.1126/science.adq1456)[45]. More DNMs were collated from the supplementary materials of Francioli et al. (https://doi.org/10.1038/ng.3292)[41] and Richter et al. (https://doi.org/10.1038/s41588-020-0652-z)[42].

The three resulting subsets (early mosaic, late mosaic and DNMs) aggregated from the studies mentioned above have been deposited in a Zenodo repository available at https://zenodo.org/records/15846952[139]. Genomic window definitions used for our main analyses are also available in the same repository. The rest of the supplementary data are available directly as Supplementary Data Files. All figure data is available as a Source Data file. Source data are provided with this paper.

## Code availability

The code we developed for the main analyses presented in this manuscript is available under a GNU General Public License v3.0 at https://doi.org/10.5281/zenodo.17226120 and as Supplementary Software 1[140].

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

## Acknowledgements

We thank the members of the Weghorn Group for helpful discussions and suggestions, especially Miguel Rodriguez Galindo. We thank Bo Xia and Itai Yanai for providing access to the curated male germline expression patterns. This publication used data from the UK Biobank under the application ID 81963. We thank Silvia Bonas Guarch and Federico Billeci for their support with UK Biobank data analysis. We also thank the reviewers for their thoughtful and constructive feedback. We acknowledge support of the Spanish Ministry of Science and Innovation through the Centro de Excelencia Severo Ochoa (CEX2020-001049-S, MCIN/AEI /10.13039/501100011033), and the Generalitat de Catalunya through the CERCA programme (D.W.). This work was supported by the Spanish Ministry of Science and Innovation (grants PGC2018-100941-A-I00 and PID2021-128976NB-I00 to D.W.) and through the Formación de Personal Investigador (FPI) programme (grant PRE2021-097261 to M.C.G.). This paper has further received funding from the European Union's Horizon 2020 MSCA postdoctoral COFUND programme (grant agreement INTREPiD No. 754422 to D.C.).

## Author contributions

Conceptualisation: D.C., D.W., data curation: M.C.G., D.C., C.S.C., D.W., formal analysis: M.C.G., D.C., C.S.C., V.S., D.W., methodology: M.C.G., D.C., C.S.C., V.S., D.W., supervision: D.W., writing: M.C.G., D.C., C.S.C., V.S., D.W.

## Competing interests

The authors declare no competing interests.
