## [Peer Review File · Nature Communications]

Transcription start sites experience a high influx of heritable variants fueled by early development

Corresponding Author: Dr Donate Weghorn

Version 0:

Reviewer comments:

Reviewer #1

(Remarks to the Author)

This study describes the discovery of extremely rare germline/early development mutational process(es) driving a mutational hotspot at transcription start sites (TSS). Mutational hotspots at TSS had previously been described in the genomes of several cancer types. Using public data for rare germline mutations, de novo mutations, and somatic mutations, the authors examined the frequency of observed mutations against the background mutation rate. The authors further took a number of steps to elucidate the mechanism associated with these hotspots, including the use of mutational signatures and regression analysis against promoter features. The authors go on to show that the mutational process driving these hotspots is also present in disease-associated genes, but the observation is not surprisingly eroded by purifying selection.

I enjoyed reviewing this paper. The findings are interesting and well supported by rigorous data analysis. The findings further our understanding of mutational processes in normal cells and tissues. As this is a purely computational analysis paper, it has generated new insights which may be further investigated by other experimental work in the future.

I have several comments for the authors' consideration:

- (1) Is there any evidence that divergent intergenic lncRNAs (i.e., those arising from an enhancer in a divergent manner) have stronger mutation hotspots?
- (2) Could the authors provide more discussion on why the specific location of the mutation hotspots is slightly different across ERVs, mosaic variants, and somatic mutations? In cancer, they are slightly upstream; in ERVs, they peak at the TSS; and in mosaic variants, they are downstream. There are some comparisons based on the regression analyses, but I think more discussion can be provided on the potential mechanism underlying the differences.
- (3) In the analysis of disease gene TSS, is there any evidence that purifying selection occurs most at conserved sites in TFBS motifs?
- (4) Although purifying selection is observed in disease-associated genes, are there disease-associated genes that don't have hotspots in the first place? Presumably, some disease genes would not be expressed in early development. Is it possible to separate disease-associated genes into those that are expressed or not expressed in early development?

(Remarks on code availability)

The paths to annotation files are hardcoded. The authors should provide the annotation files in a separate folder. Where possible the variants and mutations used in the study should also be made available.

Reviewer #2

(Remarks to the Author)

The authors of this article illustrate a surprising finding — extremely rare germline variants are enriched for non-CpG>TpG substitutions (and depleted for CpG>TpG substitutions) near transcription start sites (TSS). They speculate that the non-CpG>TpG enrichment arises from early mosaic mutations in population sequencing datasets, and demonstrate that early mosaic mutations are enriched for the same non-CpG>TpG signal near TSS. Finally, the authors decompose the TSS enrichment into mutation signatures, and attempt to pinpoint the molecular processes underpinning the enrichment.

Overall, I think this is a comprehensive analysis of an interesting genomic phenomenon, and uncovers a potentially novel

source of mutation rate variation near the TSS in human genomes. I have a few suggestions and critiques, mostly involving the filtering of "mosaic" mutations in external datasets.

Major comments:

My major concern has to do with the definition and filtering of "mosaic" mutations. If I understand correctly, the authors state that early mosaic mutations are likely filtered out of most *de novo* germline mutation callsets. This is because early mosaic mutations — which occur during embryonic development in a parent, as shown in Figure 2c — are present in both the parental germline and the parental soma. If a child is fertilized by a germ cell that contains the early mosaic mutation, that child will have the early mosaic mutation in every cell of their body. However, depending on how early the mutation occurred during the development of the parental embryo, the mutation may not appear to be a true germline *de novo* mutation, because it may be present at a non-negligible allele frequency in the parent's somatic cells. This makes complete sense, and I agree with the authors that these kinds of early mosaic mutations will frequently be filtered out of germline *de novo* mutation callsets (for example, the authors use Jonsson et al. 2021 [Nat. Genet.] as a source of early and late mosaic mutations in parents that are transmitted to multiple children, with varying amounts of parental evidence). However, another possibility is that a *de novo* mutation occurs during the early embryonic development of the *child*. If it occurs early enough (say, within the first few cell divisions following fertilization), the mutation will be present in a non-negligible fraction of the child's somatic cells, and will look indistinguishable from a germline *de novo* mutation that occurred in a parent's germ cells. In most two-generation germline *de novo* mutation studies, this kind of early mosaic mutation cannot be easily filtered out, but a small number of studies using multigenerational families have demonstrated that 5-15% of all apparent germline *de novo* mutations may, in fact, be early mosaic mutations in the child (Porubsky et al. 2025 [Nature], Sasani et al. 2019 [eLife]). Therefore, I suspect that some of the *de novo* mutations analyzed in this paper (from studies that weren't able to discriminate between early mosaic mutations in children and true germline DNMs) may actually be early mosaic mutations that occurred in a *child*. If so, it is a little surprising that the authors don't observe any non-CpG>TpG enrichment in the *de novo* mutation data.

I wonder if the authors would be able to recover some of the non-CpG>TpG TSS enrichment in their *de novo* mutation callset if they filtered those DNMs by allele balance. Early mosaic mutations in the child should be present at lower allele frequency than true germline *de novo* mutations that occurred in a parent's germ cells; if the authors filter for DNMs with an allele balance significantly lower than 0.5, could they recapitulate the non-CpG>TpG enrichment?

Perhaps I missed it, but I expected the authors to show the TSS enrichment patterns (like the ones presented in Figure 1) for the different classes of non-CpG>TpG mutations (e.g., enrichment plots for C>A, C>G, A>C, etc.). The authors perform mutation signature deconvolution on the ERVs near the TSS, but do they observe any mutation type-specific enrichment patterns near the TSS?

Finally, I wonder if the authors could comment on the degree to which early mosaic mutations (vs. other possible processes) contribute to the non-CpG>TpG enrichment at TSS in the ERVs. For example, if there is a ~50% enrichment of non-CpG>TpG mutations near the TSS in early mosaic mutations (see Figure 2f), what fraction of ERVs would need to be early mosaic mutations in order to fully explain the ~15% enrichment of non-CpG>TpG mutations near the TSS in the ERV datasets? This analysis is not necessary, but could be a relatively simple way to assess the contribution of early mosaic mutations to the ERV enrichment signal.

Specific comments and suggestions:

L132: In Figure 2c, why do the authors annotate the embryonic development diagram with a "twinning" event? I may be mistaken, but isn't "twinning" only relevant to the incidence of mono/di-zygotic twins? Since the authors don't use mutation data from twins in this study, it might make more sense to annotate the figure with the approximate timing of primordial germ cell specification (PGCS), and leave out the "twinning" annotation.

L137: Here, the authors compare two enrichment values: a) the enrichment of signatures in mutations near the TSS vs. signatures farther away from the TSS and b) the enrichment of signatures in early mosaic mutations vs signatures in germline DNMs. While this analysis makes sense, I was surprised that the authors compared mutation signatures rather than directly comparing mutation spectra. For example, the authors could compute the cosine distance between the spectra of ERVs near the TSS to the spectra of mosaic mutations, and compute the distance between the ERV spectra near the TSS to the spectra of germline *de novo* mutations. Then, they could use a permutation test to determine whether the ERV-TSS spectrum is significantly more similar to the early mosaic spectra than expected by chance. Again, this analysis is not necessary, but could be a more direct way to assess the similarities of these mutation groups.

L176: A minor comment, but in Figure 3, I expected to see the same "regressors" in both panels of the plot (and I expected the same regressors to be oriented on the same y-axis ticks). That way, we could directly compare the effect sizes of each regressor on the up- or down-stream ERV enrichment. The figure may be too messy, but could the authors show all of the regressors in both panels of the plot? Perhaps the non-significant interactions could be shaded in light grey, and the significant interactions could use the current color scheme? That way, we could more easily compare the significance of each regressor in a visually consistent way.

(Remarks on code availability)

I am unable to access the "TSS_hypermutable" GitHub repository described in the paper. Perhaps the repository needs to be made public?

Version 1:

Reviewer comments:

Reviewer #1

(Remarks to the Author)

The authors have addressed all my concerns.

(Remarks on code availability)

An example is now provided which will make it easier to navigate the code.

Reviewer #2

(Remarks to the Author)

I thank the authors for their detailed responses to my comments and questions. I believe they have addressed all of my outstanding concerns.

(Remarks on code availability)

REVIEWER COMMENTS

Reviewer #1 (Remarks to the Author):

This study describes the discovery of extremely rare germline/early development mutational process(es) driving a mutational hotspot at transcription start sites (TSS). Mutational hotspots at TSS had previously been described in the genomes of several cancer types. Using public data for rare germline mutations, de novo mutations, and somatic mutations, the authors examined the frequency of observed mutations against the background mutation rate. The authors further took a number of steps to elucidate the mechanism associated with these hotspots, including the use of mutational signatures and regression analysis against promoter features. The authors go on to show that the mutational process driving these hotspots is also present in disease-associated genes, but the observation is not surprisingly eroded by purifying selection.

I enjoyed reviewing this paper. The findings are interesting and well supported by rigorous data analysis. The findings further our understanding of mutational processes in normal cells and tissues. As this is a purely computational analysis paper, it has generated new insights which may be further investigated by other experimental work in the future.

We thank the reviewer for their encouraging remarks and insightful comments. We hope that we have been able to address them to the reviewer's satisfaction in what follows.

I have several comments for the authors' consideration:

(1) Is there any evidence that divergent intergenic lncRNAs (i.e., those arising from an enhancer in a divergent manner) have stronger mutation hotspots?

Following the FANTOM5 annotations (Hon et al., Nature, 2017), we ran the mutation density analysis across genomic windows separately for two mutually exclusive types of lncRNAs: (1) 3991 divergent-from-mRNAs lncRNAs and (2) 8454 intergenic lncRNAs, with results shown in Extended Data Figures 3 and 4, respectively. We do not define a group of lncRNAs that is considered both divergent and intergenic. However, there are lncRNAs at enhancers that are divergent from mRNAs (i.e., not considered intergenic). This latter category makes up 4% of our divergent-from-mRNAs set of 3991 lncRNAs, while the majority (87%) are located at promoters. As a whole, the set of lncRNAs that are divergent from mRNAs shows a ~33% excess in the 1 kbp upstream and a ~14% excess downstream of the TSS (Extended Data Figure 3), both larger than or equal to what is observed for protein-coding genes. This is compatible with the strong association of the protein-coding hotspot with divergent transcription (Figure 3a).

We hope this answers the reviewer's question and apologise for the limited information provided about the different types of lncRNAs present in the dataset. We adapted the Methods section to more clearly differentiate between them.

(2) Could the authors provide more discussion on why the specific location of the mutation hotspots is slightly different across ERVs, mosaic variants, and somatic mutations? In cancer, they are slightly upstream; in ERVs, they peak at the TSS; and in mosaic variants, they are downstream. There are some comparisons based on the regression analyses, but I think more discussion can be provided on the potential mechanism underlying the differences.

In the case of cancer, mutational hotspots upstream of the TSS in active promoter regions have been previously described. Our results primarily recapitulate this trend, which mechanistically was largely ascribed to the impairment of nucleotide excision repair (NER; Perera et al., Nature, 2016). NER is particularly important for the repair of bulky DNA adducts, which in turn are the result of several exogenous mutagenic insults, e.g., from UV irradiation or tobacco smoke. As the human germline is protected from such causes of DNA damage, we did not expect germline variants to be driven by similar mechanisms.

In line with this, our findings implicate divergent transcription and somatic DSBs as the primary causes of the ERV hotspot upstream of the TSS. While the regression results suggest a late origin of these mutations (i.e., shifted more towards the actual germline), we cannot rule out that a subset of these mutations arise during development, as mosaic variants cannot be removed completely from DNM data. At the same time, we observe no significant excess in early mosaic variants upstream of the TSS, suggesting that those variants are primarily driven by directly transcription-associated processes.

Consistent with this, the downstream TSS hotspot observed in ERVs appears to result primarily from mutations arising early in embryonic development. This is supported by the presence of an even stronger hotspot in early mosaic variants and its absence in *de novo* mutations (DNMs), with mosaic variants being underrepresented among DNMs. Our regression analysis further suggests that the main mechanisms underlying downstream TSS mutagenesis are mitotic double-strand breaks (DSBs), potentially occurring during developmental cell divisions or as a consequence of transcription, along with directly transcription-associated processes such as RNA polymerase II stalling and R-loop formation (see also response to comment #4 for a detailed analysis of how early developmental gene expression influences TSS mutagenesis). Although the pan-cancer dataset does not show a similar downstream TSS hotspot or associations with the same genomic and epigenetic regressors, we cannot exclude that certain cancer types which show a downstream excess might be affected by similar mechanisms (e.g., bladder, ovary, esophagus; Extended Data Figure 1).

While DSBs are well-known drivers of mutagenesis, the actual mechanisms by which RNAP II stalling may cause mutations are complex. It has been reported that stalled polymerases can prevent damage repair by directly blocking DNA access of the repair machinery, promote transcription-replication collisions, induce double-strand breaks and create R-loops (Lans et al., Nat. Rev. Mol. Cell Biol., 2019; Dellino et al., Nat. Genet. 2019). R-loops, in turn, consist of a DNA-RNA hybrid containing single-stranded DNA that is prone to causing DSBs and APOBEC-associated mutagenesis (McCann et al., Nat. Genet., 2023).

To further address the reviewer’s comment, we have added more discussion of the similarities and differences between the three datasets to the main text.

(3) In the analysis of disease gene TSS, is there any evidence that purifying selection occurs most at conserved sites in TFBS motifs?

We found that negative selection erodes the hotspot at the TSS on average across all analysed protein-coding genes (Figure 5a), but we did not separate genes by the conservation status of any TSS-adjacent TFBS motifs. We expect a decreased number of common TSS variants in (disease) genes with highly conserved TFBS motifs relative to those without, as these variants represent the remaining mutations after the action of negative selection. We thought this was a nice sanity check to support the notion that selection acts on variants contributing to the hotspot and conducted a new small analysis to address this.

Briefly, we obtained TFBS conservation data across the human genome from a UCSC genome browser track described at <https://genome.ucsc.edu/cgi-bin/hgTrackUi?g=tfbsConsSites&db=hg19> (Liu et al., Genomics, Proteomics, and Bioinformatics, 2008). The first windows of up to 1000 bp downstream and upstream of the TSS of each gene included in the disease analysis were intersected with the TFBS data to obtain the sum of conservation scores at each TSS. Genes with a summed conservation score greater than zero were considered as having conserved TFBSs at their TSSs. To estimate mutation densities, we followed a similar procedure as described in the “TSS Mutational Enrichment of Disease Phenotype Genes” section of the Methods, but separately calculated the statistics for ERVs and common variants while keeping the separation between genes with conserved TFBSs at the TSS and those without.

In the figure above, we have computed the ratio of mutation densities at the TSS and in the rest of the regions we analyse. The observed and expected number of mutations were pooled from all disease genes in each TFBS group. We observe that there is a strong action of negative selection at the TSSs of the 2055 genes with conserved TFBSs (“yes”). These genes show a ~10% depletion of common variants in their TSSs compared to the rest of the regions while showing an excess of ~10% for ERVs. Looking at the 495 disease genes with non-conserved TFBSs in their TSSs (“no”), we see a similar excess of ERVs, but no depletion of common variants. This analysis indicates that conserved TFBSs can indeed explain, at least partially, the selection signal observed at the TSS.

(4) Although purifying selection is observed in disease-associated genes, are there disease-associated genes that don't have hotspots in the first place? Presumably, some disease genes would not be expressed in early development. Is it possible to separate disease-associated genes into those that are expressed or not expressed in early development?

In total, we found 5420 out of 8411 annotated disease gene sets not to be significantly associated with a mutation excess at the TSS compared to further away. We note, however, that this does not mean that all of the 5420 diseases are truly not associated with the TSS hotspot, as we might be statistically underpowered to detect some additional true associations.

Hypothesising that the degree of relative TSS mutation enrichment in disease genes may be a function of early development gene expression makes a lot of sense given the corpus of results we present in the rest of the manuscript. We thought carrying out a new analysis to test this would be very interesting. To do this, human early development single-cell RNA-seq data was obtained from Yan et al. (Nature Structural & Molecular Biology, 2013). We took the mean RPKM expression values across replicates given in Supplementary Table 1 of the cited paper and averaged them across genes belonging to each of the HPO collections we used in our disease analysis. We then proceeded to compute the correlation between the mean expression and the pooled mutation enrichment in the TSS relative to the rest of the regions of interest (the latter was computed exactly as described in the “TSS Mutational Enrichment of Disease Phenotype Genes” section of the Methods).

In the density histogram above, each data point represents an HPO collection and the red curve a LOESS trend fit (data points are binned to enhance visibility). To assess the significance of the correlation, we were unable to use standard parametric approaches, as points are not independent due to multiple genes appearing across many HPO terms. Instead, we permuted the HPO collection memberships across the analysed genes 10,000 times to generate random collections of genes with the same size as the original collections, allowing for each gene to be repeated across collections as many times as it is actually repeated in the real HPO. For each random realization, we calculated the mean expression and pooled mutation density ratio followed by the correlation coefficient between the two across collections. We computed an empirical p-value by calculating how many permutations had a correlation greater than or equal to the observed correlation, giving $p < 10^{-4}$. We conclude that early developmental gene expression can indeed partially explain the TSS hotspot phenomenon and that this correlation is also observed at the disease collection level.

The source publication of the expression data we are using profiles human development transcriptional states at distinct stages. To decide which stage to use for the previous analysis, we analysed the correlation between expression and TSS mutational enrichment across genes for each stage and picked the one with maximum correlation (morulae stage). The details can be appreciated in the following plot:

a Log10(RPKM + 1) Pearson correlation matrix**b Log10(RPKM + 1) & mutation density ratio (TSS / far)**
In panel **a**, we identify two major clusters of transcriptional states based on their Pearson correlation across genes (all correlations $p_{adj} \ll 10^{-16}$). Note that all stages are chronologically ordered except for epiblast and primitive endoderm (P. endoderm), which arise roughly at the same time after the trophectoderm (Arthur & Chazaud., Cellular and Molecular Life Sciences, 2014). In panel **b**, we see that the highest gene expression correlations with TSS mutational enrichment are found within the second cluster, which is composed of stages between 8-cell and epiblast (all individual correlations shown in this panel $p_{adj} \ll 10^{-16}$). As morulae had the highest correlation, we tested whether its correlation coefficient was different from the ones computed for all other stages. This analysis indicates that the correlation of morulae expression with TSS mutational enrichment is significantly higher compared to most stages in the first cluster but not clearly different from correlations in the second cluster (brackets showing adjusted p-values above the bars). Altogether, this suggests that any transcription-related damage potentially experienced during early development in protein-coding TSSs would intensify around the 8-cell stage and possibly continue throughout the morulae stage and until the time of differentiation of the trophectoderm. The association then seems to become milder by the time that the primitive endoderm and epiblast arise from the inner cell mass. This putative increase in damage coincides with the major transcriptional shift experienced between the 4-cell and 8-cell stages.

We think that this analysis presents a valuable piece of information regarding the impact of early developmental transcription on TSS mutagenesis and have added it to the manuscript as the new Extended Data Figure 15, with the methodological details written in the “Correlation Between Developmental Gene Expression and TSS Mutational Enrichment” section of the Methods.

Reviewer #1 (Remarks on code availability):

The paths to annotation files are hardcoded. The authors should provide the annotation files in a separate folder. Where possible the variants and mutations used in the study should also be made available.

The code has been updated with the suggestions and now it is easier to analyse any given dataset with the right formatting (examples are provided). We have also deposited the annotations for the genomic windows we use in the main analysis of the paper here:

https://zenodo.org/records/15846952?token=eyJhbGciOiJIUzUxMiJ9.eyJpZCI6IjZmM4YmY1LWE4Y2MtNGE2ZS04NDNjLWVhNzZlZmYyYWYzYyIsImRhdGEiOnt9LCJyYW5kb20iOiI4ZjhhZjhmYzJmNW11ZDZlYmNlYzIxMmRhOWFiMDk0NyJ9.8lnW4YCSyvQla922OZB4X9fYUfYmj4Sy2V2PAo2d2T0_4O_b1z_BuXWQVMtIFeyPo0So9ZUTedJ_rRizzbC9tg

The mutations belonging to the meta-cohorts we have assembled are also available in the same link, as well as in Supplementary Tables 9-11, now added to the manuscript. Other mutation datasets are possible to already directly access elsewhere so we have not included them due to their big size.

Reviewer #2 (Remarks to the Author):

The authors of this article illustrate a surprising finding — extremely rare germline variants are enriched for non-CpG>TpG substitutions (and depleted for CpG>TpG substitutions) near transcription start sites (TSS). They speculate that the non-CpG>TpG enrichment arises from early mosaic mutations in population sequencing datasets, and demonstrate that early mosaic mutations are enriched for the same non-CpG>TpG signal near TSS. Finally, the authors decompose the TSS enrichment into mutation signatures, and attempt to pinpoint the molecular processes underpinning the enrichment.

Overall, I think this is a comprehensive analysis of an interesting genomic phenomenon, and uncovers a potentially novel source of mutation rate variation near the TSS in human genomes. I have a few suggestions and critiques, mostly involving the filtering of "mosaic" mutations in external datasets.

We thank the reviewer for their thorough assessment of our manuscript and their interesting suggestions. Please find our responses below.

Major comments:

1. My major concern has to do with the definition and filtering of "mosaic" mutations. If I understand correctly, the authors state that early mosaic mutations are likely filtered out of most *de novo* germline mutation callsets. This is because early mosaic mutations — which occur during embryonic development in a parent, as shown in Figure 2c — are present in both the parental germline and the parental soma. If a child is fertilized by a germ cell that contains the early mosaic mutation, that child will have the early mosaic mutation in every cell of their body. However, depending on how early the mutation occurred during the development of the parental embryo, the mutation may not appear to be a true germline *de novo* mutation, because it may be present at a non-negligible allele frequency in the parent's somatic cells. This makes complete sense, and I agree with the authors that these kinds of early mosaic mutations will frequently be filtered out of germline *de novo* mutation callsets (for example, the authors use Jonsson et al. 2021 [Nat. Genet.] as a source of early and late mosaic mutations in parents that are transmitted to multiple children, with varying amounts of parental evidence). However, another possibility is that a *de novo* mutation occurs during the early embryonic development of the *child*. If it occurs early enough (say, within the first few cell divisions following fertilization), the mutation will be present in a non-negligible fraction of the child's somatic cells, and will look indistinguishable from a germline *de novo* mutation that occurred in a parent's germ cells. In most two-generation germline *de novo* mutation studies, this kind of early mosaic mutation cannot be easily filtered out, but a small number of studies using multigenerational families have demonstrated that 5-15% of all apparent germline *de novo* mutations may, in fact, be early mosaic mutations in the child (Porubsky et al. 2025 [Nature], Sasani et al. 2019 [eLife]). Therefore, I suspect that some of the *de novo* mutations analyzed in this paper (from studies that weren't able to discriminate between

early mosaic mutations in children and true germline DNMs) may actually be early mosaic mutations that occurred in a *child*. If so, it is a little surprising that the authors don't observe any non-CpG>TpG enrichment in the *de novo* mutation data.

We fully agree with the reviewer that some variants included in the DNM dataset are early mosaic variants that happened in the child, as it would be impossible to distinguish them from true germline (i.e., single-gamete) variants coming from the parents given the setup used in most of the family sequencing experiments from which we obtained data. At the same time, the amount of early mosaic variants relative to all other variants such as true germline and late mosaic variants should be higher in ERVs compared to the DNM dataset we assembled. The reason is that ERV datasets contain the following variants:

- Early mosaic mutations at leaf nodes: for the people at the bottom of the phylogeny, early mosaic mutations appear in their blood but their germline mutations are never seen as they do not have children where such mutations can be probed.
- Early mosaic mutations in the parents of leaf nodes.
- Germline mutations in the parents of leaf nodes.
- “Ancestral” early mosaic mutations: these are the mutations that happened in grandparents or higher levels of the phylogeny with respect to the leaf nodes (up to a maximum population allele frequency of 0.01% and the maximum number of generations contributing mutations to ERVs).
- “Ancestral” germline mutations.

While in the DNMs dataset there are only:

- Early mosaic mutations in the children (not present in 2nd generation DNMs obtained from Sasani et al., eLife, 2019).
- Germline mutations in the parents.

In other words, in the parental generation of the leaf nodes, there is an additional “dose” of early mosaic mutations in ERVs compared to family sequencing data. We think that this difference is responsible for the observed difference between ERVs (1.14 ± 0.00) and DNMs (1.04 ± 0.08) in the first 1000 bp downstream of the TSS in Figure 2a. To make this point clear in the manuscript, we added a note on the contribution of the child's mosaic variants to the main text. In the answer to comment #4, we further discuss what would be the fraction of early mosaic variants in ERVs that could explain the enrichment we observe and how it compares to the fraction in DNMs.

2. I wonder if the authors would be able to recover some of the non-CpG>TpG TSS enrichment in their *de novo* mutation callset if they filtered those DNMs by allele balance. Early mosaic mutations in the child should be present at lower allele frequency than true germline *de novo* mutations that occurred in a parent's germ cells; if the authors filter for DNMs with an allele balance significantly lower than 0.5, could they recapitulate the non-CpG>TpG enrichment?

We thank the reviewer for this interesting idea. Among our ten DNM datasets, two provided the reference and alternate read counts for the mutations observed in the child's blood (Jonsson et al., Nature, 2017; Jonsson et al., Nat. Genet., 2021; see histograms below). We noticed that the distributions exhibited a positive skew, suggesting that low-*VAF* variants had already been filtered by the authors. Nevertheless, we isolated 11830 variants with $VAF < 0.4$ and re-ran our analysis pipeline on this restricted set of putative early mosaic variants in the child. However, somewhat expectedly, this analysis did not produce a significant excess of mutations at the TSS (or anywhere else).

- Perhaps I missed it, but I expected the authors to show the TSS enrichment patterns (like the ones presented in Figure 1) for the different classes of non-CpG>TpG mutations (e.g., enrichment plots for C>A, C>G, A>C, etc.). The authors perform mutation signature deconvolution on the ERVs near the TSS, but do they observe any mutation type-specific enrichment patterns near the TSS?

In Figure 4e and Extended Data Figure 18 (formerly 16), we show six panels with estimates of mutation densities subdivided by mononucleotide substitution type and also by the strand in which mutations are counted relative to the directions of transcription and replication. At the 100-bp level, all 12 mutation types show an excess at the TSS (Extended Data Figure 18).

- Finally, I wonder if the authors could comment on the degree to which early mosaic mutations (vs. other possible processes) contribute to the non-CpG>TpG enrichment at TSS in the ERVs. For example, if there is a ~50% enrichment of non-CpG>TpG mutations near the TSS in early mosaic mutations (see Figure 2f), what fraction of ERVs would need to be early mosaic mutations in order to fully explain the ~15% enrichment of non-CpG>TpG mutations near the TSS in the ERV datasets? This analysis is not necessary, but could be a relatively simple way to assess the contribution of early mosaic mutations to the ERV enrichment signal.

We think going through this calculation is a really nice suggestion and we have estimated a few numbers to address it. First, we note that the excess of ERVs in the first window of 1000 bp (1 kbp) downstream of the TSS can be expressed as:

$$r_e = \frac{np_e f + n(1 - p_e)\ell/L}{n\ell/L}$$

where we assume that non-mosaic variants are distributed uniformly across the genome (we account for the fact that early-replicating regions are generally less mutable compared to late-replicating regions and that μ is a compositional observable in the calculations of the ratios). Here, n is the total number of mutations in the ERV dataset (within all our regions of interest), p_e is the proportion of mutations in the whole ERV dataset that are early mosaic, f is the fraction of early mosaic mutations that fall into the 1 kbp window downstream of the TSS, ℓ is the total length of genomic regions in the 1 kbp TSS window (summed over all genes) and L is the total length of all genomic regions considered. f can be estimated by simply calculating the proportion of total mutations that are at the TSS for the dataset that was collated to contain only early mosaic variants. Rewriting the equation lets us solve for p_e :

$$p_e = \frac{r_e - 1}{fL/\ell - 1}$$

Plugging $\ell/L = 0.015$, $f = 0.024$ and $r_e = 1.14$ into this equation indicates that the proportion of early mosaic variants out of all gnomAD ERVs that is compatible with the enrichment at the TSS is $p_e \approx 22\%$. We can also estimate p_e using the observed excess of early mosaic variants in the 1 kbp downstream of the TSS, which is just

$$r_m = fL/\ell$$

Substituting $r_m = 1.52/0.9 = 1.69$ in the previous formula yields $p_e \approx 20\%$, compatible with the first estimate.

Assuming that ERVs capture one generation and ignoring the contribution of purely somatic mutations in the blood of the probands, we can further derive an expression for the proportion of early mosaic variants for DNMs, p_d , and ERVs in terms of the number of mosaic variants, m , and true germline variants, g , per individual:

$$p_d = \frac{m}{m + g} \quad \text{and} \quad p_e = \frac{2m}{2m + g}$$

Using $p_e \approx 21\%$, this yields $p_d \approx 12\%$, in very good agreement with the lower bound on p_d of 9.5% estimated in Sasani et al. (eLife, 2019).

Of course, even though they are rare, ERVs were not all generated in the last generation and so the true p_d might be larger (but always smaller than p_e). The full expression for p_e as a function of m and g would require a weighted sum over the ERV allele frequencies, integrating over the distribution of allele age for each allele frequency, for which no closed form exists even when assuming constant population size and neutral evolution (Griffiths and Tavaré, Commun. Statist.,

1998). However, the distribution of allele age is heavily skewed towards small ages, in line with the intuition that most singletons would have arisen very recently. So we expect the weighted sum to be dominated by recent alleles, for which the mosaic fraction is $2m/(2m + g)$ and which would therefore give a similar result as above. We also note that allele age is further reduced in expanding populations, which applies to the recent human population, and under purifying selection, which we showed is affecting ERVs to some extent (e.g., $dN/dS \approx 0.82$ on protein-coding exons). This suggests that p_d would indeed be substantially lower than p_e if all evolutionary dynamics were known and factored in. Overall, even though these calculations are simplistic and rely on several assumptions, they reinforce our interpretation of the factors driving the differences between ERVs and DNMs within the first 1 kbp downstream of the TSS.

Specific comments and suggestions:

5. L132: In Figure 2c, why do the authors annotate the embryonic development diagram with a "twinning" event? I may be mistaken, but isn't "twinning" only relevant to the incidence of mono/di-zygotic twins? Since the authors don't use mutation data from twins in this study, it might make more sense to annotate the figure with the approximate timing of primordial germ cell specification (PGCS), and leave out the "twinning" annotation.

We included the twinning event in the illustrative Figure 2c because one of the datasets we are using indeed reports mutations from monozygotic twins (Jonsson, Nat. Genet., 2021). This is described in the "De Novo and Mosaic Mutation Data Sets" subsection in the Methods. We appreciate that this complicates the figure a little, but we felt it was necessary to make this additional distinction between different types of early mosaic variants.

6. L137: Here, the authors compare two enrichment values: a) the enrichment of signatures in mutations near the TSS vs. signatures farther away from the TSS and b) the enrichment of signatures in early mosaic mutations vs signatures in germline DNMs. While this analysis makes sense, I was surprised that the authors compared mutation signatures rather than directly comparing mutation spectra. For example, the authors could compute the cosine distance between the spectra of ERVs near the TSS to the spectra of mosaic mutations, and compute the distance between the ERV spectra near the TSS to the spectra of germline de novo mutations. Then, they could use a permutation test to determine whether the ERV-TSS spectrum is significantly more similar to the early mosaic spectra than expected by chance. Again, this analysis is not necessary, but could be a more direct way to assess the similarities of these mutation groups.

We think this is a nice suggestion and decided to carry out the analysis to at least compare ERVs and DNMs. While it would be nice to also compare ERVs and early mosaic mutations, we anticipated the test would always be non-significant as we are severely underpowered to detect differences in mutational spectra given the low number of early mosaic mutations in the first window downstream of the TSS.

For the ERV-DNM comparison, we aimed for using something similar to the Aggregate Mutation Spectrum Distance (AMSD) method (Hart et al., bioRxiv, 2025). However, we were unable to use it exactly as described given that we lack the sample identifiers of each mutation in our datasets. Instead of assigning the same randomized group label to all mutations coming from the same sample, we have to independently permute the labels of each mutation separately and take this as our best approximation to the AMSD method. This yields cosine distances between mutation profiles drawn from identical distributions that are mixtures of DNM and ERV profiles. We used these cosine distances to construct null distributions of the lower bounds of the cosine distances between the ERV and DNM spectra in the first 1 kbp window downstream and upstream of the TSS.

Using these null distributions (histograms in the plot above), an empirical p-value was computed for the observed cosine distances (red dotted lines). The average randomized cosine distance upstream was larger than downstream, likely driven by the smaller number of mutations upstream compared to downstream, which introduces larger variance into the permuted mutation patterns. To compare the effect sizes downstream and upstream, we then divided the observed cosine distance by the mean randomized distance in each case (fold changes shown at the bottom of the histograms). This revealed a greater difference between spectra downstream of the TSS than upstream. This is compatible with the hypothesis that ERVs and DNMs are more differentiated immediately downstream of the TSS compared to upstream due to a higher load of early mosaic mutations in ERVs downstream, which are largely absent in the DNMs.

7. L176: A minor comment, but in Figure 3, I expected to see the same "regressors" in both panels of the plot (and I expected the same regressors to be oriented on the same y-axis ticks). That way, we could directly compare the effect sizes of each regressor on the up- or down-stream ERV enrichment. The figure may be too messy, but could the authors show all of the regressors in both panels of the plot? Perhaps the non-significant interactions could be shaded in light grey, and the significant interactions could use the

current color scheme? That way, we could more easily compare the significance of each regressor in a visually consistent way.

We agree with the reviewer that it would give a more complete picture to show all regressors in Figure 3. We decided to restrict to only the significant regressors in the figure for the sake of focusing on the relevant factors. However, in order to address the reviewer's comment, we have now added Extended Data Figure 14 showing all regressors, based on the data provided in Supplementary Table 2.

Reviewer #2 (Remarks on code availability):

I am unable to access the "TSS_hypermutable" GitHub repository described in the paper. Perhaps the repository needs to be made public?

The repository is private for the time being but we provide an exact copy of the code in the repository to the reviewers as a zip file together with the resubmission, just like we did with the original submission. The repository will be made public upon publication of the manuscript.